# Sharing is Sabotaging: Cross-Client Poisoning Attacks on Federated Graph Learning

## Abstract

Federated Graph Learning (FGL) enables collaborative training of Graph Neural Networks (GNNs) without sharing raw data. While prior work has focused on privacy leakage risks, we reveal a more direct and structurally embedded threat: a gray-box poisoning attack that manipulates shared neighbor representations during training. Specifically, we show that auxiliary-information-sharing frameworks, which collaboratively train node generators, create novel and powerful attack surfaces. A malicious client can poison shared gradients during generator updates, causing benign clients to unknowingly incorporate compromised synthetic nodes into their local subgraphs. These poisoned nodes propagate corrupted signals during training, resulting in persistent model degradation. We formalize this threat model, propose a stealthy optimization-based attack, and demonstrate its effectiveness in degrading model performance while evading standard defenses. The attack is highly stealthy, model-agnostic, and remains effective across diverse datasets, partition strategies, and numbers of clients. Our findings highlight a critical vulnerability in FGL systems and underscore the urgent need for dedicated robustness mechanisms against attacks targeting the data-generation layer.

## 1 Introduction

Graph Neural Networks (GNNs) have emerged as a powerful paradigm for learning from graph-structured data, leveraging topological dependencies through message passing between nodes (Corso et al., 2024). In practice, however, many high-value graph datasets are inherently distributed across multiple organizations due to privacy regulations or institutional barriers—think hospital networks, financial consortia, or social platforms. Federated Graph Learning (FGL) (Zheng et al., 2021; Xie et al., 2021; Baek et al., 2023) address this problem, enabling collaborative GNN model training across clients *without* requiring each party to share its raw graph data. Instead, clients share local model parameters or gradients to jointly train a global model.

Unfortunately, the fact that graph data is not shared across clients leads to the inevitable loss of inter-subgraph connections. Since these missing interconnections cannot be taken into account during collaborative training, this results in suboptimal model performance, as potentially critical topological dependencies remain unexploited. To address this challenge, dedicated FGL frameworks have been proposed. They strategically share during training auxiliary graph information such as aggregated node features (Yao et al., 2023) or node embeddings (Wu et al., 2021; Zhang et al., 2021). Crucially, a prevalent solution involves training shared auxiliary components, such as node generators (Zhang et al., 2021; Peng et al., 2022; Zhang et al., 2024), spectral encoders (Tan et al., 2024), or global synthetic data synthesizers (Kim et al., 2025), to reconstruct missing inter-client structural information. Among these, a representative work **FedSage+** (Zhang et al., 2021) tackles missing cross-subgraph edges by introducing generators collaboratively trained with the help of shared auxiliary data to approximate remote neighbors for each client. This significantly boosts model performance.

While such *auxiliary-info-sharing* FGL approaches represent an important improvement over basic federated training, they introduce a profound new vulnerability. By relaxing the information-sharing bottleneck, auxiliary-info-sharing FGL methods create new attack surfaces. In particular, Yao et al. (2024) has demonstrated a privacy-leakage risk as it is possible to recover sensitive attributes through gradient inversion.

In this paper, we reveal that beyond the conventional risk of passive privacy leakage, auxiliary-information-sharing FGL methods, represented by FedSage+, introduce a significantly more severe and active threat of **Indirect Gradient Poisoning Attack (IGPA)**. The very mechanism enabling performance gains (i.e., auxiliary information sharing) creates direct vectors for malicious clients to manipulate the model. Specifically, FedSage+ equips each client with a node generator—called NeighGen—that creates synthetic nodes to augment local subgraphs, followed by standard federated training of a global GNN model. Crucially, clients collaboratively train their generators by exchanging gradient information during federated updates. This establishes **a critical attack vector**: a malicious client can *poison shared gradients to stealthily control the properties of synthetic nodes* generated across the system. These compromised nodes then propagate poisoned information through local subgraphs during GNN training, ultimately degrading global model performance. Importantly, IGPA functions as an *indirect data manipulation attack*: rather than directly altering raw data, the adversary manipulates the shared generator gradients so that benign clients themselves instantiate poisoned training data.

The proposed IGPA has several distinctive and concerning features compared to established FL attacks: (1) **Novelty**. Unlike other attacks that target federated model parameters or directly tamper with training data, IGPA uses the generator-sharing process to alter the data structure underlying benign clients. This two-stage influence is likely to evade detection by server-side robust aggregation or other model-centric defenses; (2) **Stealthiness**. Malicious synthetic nodes are indistinguishable from legitimate ones since their generation process is core to the framework; conventional anomaly detection at the server is ineffective, given the shared component is graph-valued and not observable at the central aggregator; (3) **Amplification**. By corrupting data generators rather than weights, IGPA persists and compounds over rounds, gradually redirecting clients' learning trajectory.

Despite its potent threat profile, executing IGPA faces inherent challenges. An adversary must: (i) operate with only partial victim information (e.g., node embeddings), complicating the generation of targeted adversarial gradients across diverse victims; (ii) effectively exploit gradient propagation to manipulate the victim's training process; and (iii) maintain stealthiness to evade detection throughout the federated learning process. These challenges raise two critical questions: (1) How can an adversary overcome these obstacles to launch a successful IGPA? (2) What is the ultimate impact on the global model's performance and robustness once these challenges are surmounted?

To answer these questions, this paper makes the following contributions:

- We formalize a novel gray-box poisoning threat model for auxiliary-info-sharing FGL, wherein a malicious client poisons shared graph-related gradients to compromise the global model—*without* requiring access to other clients' raw graph data or local parameters.
- We propose IGPA, a gradient-guided poisoning method against FGL. By approximating benign update directions and injecting malicious signals, the attack effectively reduces node classification accuracy while maintaining stealth.
- We examine the stealthiness guarantees of IGPA and, with a focus on FedSage+, empirically demonstrate its effects across a variety of datasets, where we observe a substantial decline in model performance for benign clients.

## 2 BACKGROUND AND PROBLEM DEFINITION

### 2.1 FEDERATED GRAPH LEARNING

In this work, we focus on node classification as the primary learning task within the federated graph learning (FGL) framework. Consider a global graph $\mathcal{G} = \{\mathbb{V}, \mathbb{E}, \boldsymbol{X}\}$, where $\mathbb{V}$ is the set of nodes, $\mathbb{E}$ the set of edges, and each node $v \in \mathbb{V}$ is associated with a $d$-dimensional feature vector $\boldsymbol{x}_v$. In the FL setting, this graph is partitioned across $M$ clients $\{D_1, \ldots, D_M\}$. Each client $C_i$ holds a subgraph $\mathcal{G}_i = \{\mathbb{V}_i, \mathbb{E}_i, \boldsymbol{X}_i\}$ such that the node sets across clients are disjoint. Each node has a one-hot label vector used for downstream tasks like node classification. The ego-graph of a node $v$ is denoted as $\mathcal{G}(v)$, and training samples are drawn as $(\mathcal{G}(v), y_v)$.

FGL aims to collaboratively learn a global node classifier $\mathcal{F}$ by minimizing the average loss across all clients. To this end, Xie et al. (2021) introduced FedSage—a combine graph mining model GraphSage (Hamilton et al., 2018) with well-known FedAvg framework (McMahan et al., 2017).

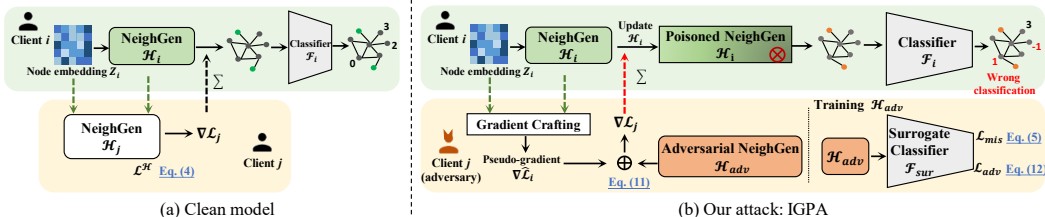

(a) Clean model  (b) Our attack: IGPA

Figure 1: Clean model and the overview of IGPA

FedSage's main drawback is that it cannot fully model the entire structure of the graph because each client only accesses its own subgraph. This leads to the "missing neighbor problem", i.e., missing connections between subgraphs. Because of it, local models are biased and less able to generalize, and accuracy is lower overall (as compared to centralized models with access to the complete graph).

## 2.2 THE TARGET MODEL

The target of the attack considered in this paper is FedSage+ which address the problem of Fed-Sage by adding a missing neighbor generation mechanism called NeighGen as shown in 1(a). The NeighGen module consists of two primary components: a $K$-layer GraphSAGE encoder and a generator $\mathcal{H}$. For each node $v \in \mathbb{V}_i$ of the input subgraph $\mathcal{G}_i$, the encoder computes node embeddings $\boldsymbol{Z}_i = \{\boldsymbol{z}_v | v \in \mathbb{V}_i\}$, where $\boldsymbol{z}_v \in \mathbb{R}^{d_z}$. The generator $\mathcal{H}$ aims to generate missing neighbors for the nodes in $\mathcal{G}_i$ based on the embeddings learned by the encoder. $\mathcal{H}$ is composed of two sub-modules: a linear regression model $\mathcal{H}^{\text{deg}}$ that predicts the number of missing neighbors for each node, and a feature generator $\mathcal{H}^{\text{feat}}$ that outputs the features for the missing neighbors.

When training $\mathcal{H}$, we randomly hold out part of its nodes $\mathbb{V}_i^h \subset \mathbb{V}_i$ and all links involving them to form an impaired subgraph, denoted as $\bar{\mathcal{G}}_i = \{\bar{\mathbb{V}}_i, \bar{\mathbb{E}}_i, \bar{\boldsymbol{X}}_i\}$, where $\bar{\mathbb{V}}_i = \mathbb{V}_i \setminus \mathbb{V}_i^h$. Client $C_i$ then optimizes $\mathcal{H}^{\text{deg}}$ by minimizing a smooth $l_1$ loss (Girshick, 2015) between the predicted and actual number of neighbors for each remaining node in $\bar{\mathbb{V}}_i$. Meanwhile, $\mathcal{H}^{\text{feat}}$ is trained using a greedy feature-matching loss: for each predicted missing neighbor of a node $v \in \bar{\mathbb{V}}_i$, the generator attempts to produce a feature vector that is as close as possible (in terms of Euclidean distance) to the most similar ground-truth feature among the held-out neighbors in $\mathbb{V}_i^h$. To jointly train NeighGen, FedSage+ adds a cross-subgraph feature reconstruction loss into $\mathcal{H}_i$ as follows:

$$\mathcal{L}_i^{\mathcal{H}} + \alpha \frac{1}{|\bar{\mathbb{V}}_i|} \sum_{v \in \bar{\mathbb{V}}_i} \sum_{p \in [\tilde{n}_v]} \sum_{j \in [M]/i} \min_{u \in \mathbb{V}_j} \left( \|\tilde{\boldsymbol{x}}_v^p - \boldsymbol{x}_u\|_2^2 \right). \tag{1}$$

In this joint training, $C_i$ sends out $\mathcal{H}_i$ and $\boldsymbol{Z}_i$. Received this information, $C_j$ computes predicted node $v$'s feature $\tilde{\boldsymbol{x}}_v^p = \mathcal{H}_i(\boldsymbol{z}_v)^p$ and the loss $\sum_{p \in [\tilde{n}_v]} \min_{u \in \mathbb{V}_j} \left( \|\tilde{\boldsymbol{x}}_v^p - \boldsymbol{x}_u\|_2^2 \right)$. Finally, $C_j$ sends the model gradient of this loss term back to $C_i$:

$$\nabla \mathcal{L}_j^{\mathcal{H}} = \frac{\partial \sum_{p \in [\tilde{n}_v]} \min_{u \in \mathbb{V}_j} \left( \|\tilde{\boldsymbol{x}}_v^p - \boldsymbol{x}_u\|_2^2 \right)}{\partial \mathcal{H}_i} \tag{2}$$

After receiving all the gradients, client $C_i$ updates its own local gradient by adding a weighted sum of the received gradients from all other clients, where each received gradient is scaled by a factor $\alpha$. Finally, $\mathcal{G}_i$ is passed through the well-trained NeighGen to obtain an altered subgraph $\mathcal{G}_i'$, and the classifier $\mathcal{F}$ will be trained on $\mathcal{G}_i'$.

## 2.3 THREAT MODEL

We consider a gray-box attacker, modeled as a malicious client in the FedSage+ framework. Similar to benign clients, the attacker has access to its local subgraph, a NeighGen and a classifier, and during training it also receives generators $\mathcal{H}$ and node embeddings $\boldsymbol{Z}$ from other clients. The attacker's objective is to degrade the global model's performance by manipulating its own local graph before contributing to collaborative training. Specifically, it injects carefully crafted malicious nodes,

edges, and attributes into its subgraph such that, after the global training procedure, the test accuracy of the resulting model is minimized. Formally, the attacker seeks to solve:

$$\frac{1}{M} \sum_i \max_{\mathcal{G}'_i} \sum_{t \in \mathbb{V}^{\text{tar}}_i} \mathbb{I}(\mathcal{F}(\phi^*; \mathcal{G}'_i(t)) \neq y_t)$$

$$\mathcal{G}' = (\mathbb{V} \cup v_{\text{inj}}, \mathbb{E} \cup e_{\text{inj}}, \boldsymbol{X} \oplus \boldsymbol{x}_{\text{inj}}) \tag{3}$$

$$\text{s.t. } \phi^* = \arg\min_{\phi} \mathcal{L}(\mathcal{F}(\phi)),$$

where $\mathbb{I}(\cdot)$ is an indicator function, $\oplus$ denotes concatenation, $\mathbb{V}_{\text{tar}}$ is the set of target nodes, and $y_t$ is the ground truth label of node $t$. The altered graph $\mathcal{G}'$ includes the injected node $v_{\text{inj}}$, its edges $e_{\text{inj}}$, and attributes $\boldsymbol{x}_{\text{inj}}$, which are optimized to maximize the number of misclassified testing nodes.

## 3 METHODOLOGY

### 3.1 ATTACK OVERVIEW

In FedSage+, each client contains a NeighGen (missing-neighbor generator) and a classifier. Since the classifier depends entirely on the mended graph produced by NeighGen, corrupting the Neigh-Gen produces a stealthy poisoning vector. Under a gray-box threat model (attacker knows its local data/model and shared generators/embeddings), IGPA proceeds by (i) training a benign surrogate classifier on the attacker's local data, (ii) optimizing an adversarial NeighGen to induce misclassification on that surrogate, and (iii) crafting pseudo-gradients that blend adversarial and substitutional information, to steer benign clients' NeighGen parameters toward the malicious NeighGen, as illustrated in Figure 1. By embedding the attack inside the generator rather than by directly altering stored graphs, the attack produces plausible NeighGen outputs that manifest as legitimate augmentations, making detection difficult while steadily degrading global classification performance.

### 3.2 SURROGATE CLASSIFIER TRAINING

In the context of adversarial graph learning, an attacker seeks to manipulate the graph structure to mislead benign clients' models. To achieve this, the attacker follows a two-phase strategy, first training a surrogate classifier to approximate the benign model's behavior and then optimizing an adversarial graph generation model, $\mathcal{H}_{\text{adv}}$, to manipulate the graph structure to confuse the surrogate.

Initially, the attacker trains a surrogate victim model by mimicking the process of a benign client. This is done by optimizing the benign NeighGen $\mathcal{H}_j$ through the minimization of a loss function that consists of two primary terms. The first term, $L_1^S$, is a smooth L1 loss that ensures the predicted feature for a given node is close to its actual feature. The second term incentivizes the model to predict neighboring nodes for missing neighbors, where the similarity between the predicted feature and the actual feature of the neighbor node is measured using the L2 norm. Specifically, the loss function is formulated as:

$$\mathcal{L}^{\mathcal{H}} = \frac{1}{|\bar{\mathbb{V}}_j|} \left( \sum_{v \in \bar{\mathbb{V}}_j} L_1^S(\widetilde{n}_v - n_v) + \sum_{v \in \bar{\mathbb{V}}_j} \sum_{p \in [\widetilde{n}_v]} \min_{u \in \mathcal{N}_{\mathcal{G}_i}(v) \cap \mathbb{V}^h_j} \left( \|\widetilde{\boldsymbol{x}}^p_v - \boldsymbol{x}_u\|^2_2 \right) \right), \tag{4}$$

where $\widetilde{\boldsymbol{x}}^p_v$ denotes the predicted feature for the $p$-th missing neighbor of node $v$, and $\mathcal{N}_{\mathcal{G}_i}(v)$ represents the neighborhood of node $v$ in the graph $\mathcal{G}_i$.

### 3.3 ADVERSARIAL NEIGHGEN OPTIMIZATION

Once the surrogate victim model $\mathcal{F}_{\text{sur}}$ is trained, the attacker proceeds to the next phase: the optimization of an adversarial NeighGen, $\mathcal{H}_{\text{adv}}$, which is designed to manipulate the graph structure in a way that maximizes the misclassification rate of the surrogate model. This is achieved by introducing an adversarial perturbation into the graph generation process. The goal of the adversarial NeighGen is to identify graph alterations that induce misclassification in the surrogate model, thereby making the benign model vulnerable to the attack.

To achieve this, the adversarial NeighGen is optimized using a misclassification-oriented loss function denoted as $\mathcal{L}_{\text{mis}}$. Here, the attacker flips the objective to encourage misclassification. Specifically, the loss is defined as the negative cross-entropy:

$$\mathcal{L}_{\text{mis}} = - \sum_{v \in \mathbb{V}_j} \mathbb{I}[y_v \neq \mathcal{F}_{\text{sur}}(v, \mathcal{G}_i)] \log \mathcal{F}_{\text{sur}}(v, \mathcal{G}_i), \tag{5}$$

where $\mathbb{I}[y_v \neq \mathcal{F}_{\text{sur}}(v, \mathcal{G}_i)]$ indicates a misclassification and $\mathcal{F}_{\text{sur}}(v, \mathcal{G}_i)$ is the output of the surrogate model. By training $\mathcal{H}_{\text{adv}}$ to generate perturbations that consistently lead to misclassification, the attacker obtains a mechanism that can later be leveraged to poison the training process of benign NeighGen models.

### 3.4 Gradient Crafting and Poisoning

The key idea of the attack is to replace benign client $C_i$'s NeighGen $\mathcal{H}_i$ with the attacker $C_j$'s adversarial NeighGen $\mathcal{H}_{\text{adv}}$. For notational simplicity, we denote $\nabla \mathcal{L}_i \equiv \nabla_{\mathcal{H}_i} \mathcal{L}^{\mathcal{H}_i}$ for any client $C_i$. Notably, the joint update of gradient of client $C_i$'s NeighGen is:

$$\nabla \mathcal{L}_i \leftarrow \nabla \mathcal{L}_i + \alpha \Big( \sum_{m \in [M]/\{i,j\}} \nabla \mathcal{L}_m + \nabla \mathcal{L}_j \Big). \tag{6}$$

To illustrate, we consider SGD optimizer, where the parameters are updated according to $\mathcal{L}_i \leftarrow \mathcal{L}_i - \eta \nabla \mathcal{L}_i$, with $\eta$ as learning rate . To let $\mathcal{H}_i = \mathcal{H}_{\text{adv}}$, client $C_j$ needs to send gradient

$$\nabla \mathcal{L}_j = \frac{1}{\alpha} \left( \frac{\mathcal{H}_i - \mathcal{H}_{\text{adv}}}{\eta} - \nabla \mathcal{L}_i \right) - \sum_{m \in [M]/\{i,j\}} \nabla \mathcal{L}_m, \tag{7}$$

to the benign client $C_i$. However, the attacker cannot access $\nabla \mathcal{L}_i$. Therefore, to calculate $\nabla \mathcal{L}_j$, the attacker must recover this gradient using other available information ($z_v$ and $\mathcal{H}_i$).

Inspired by the pseudo-label-based unsupervised learning, we first construct a pseudo-label $\hat{y}_i = \operatorname{argmax}(\mathcal{H}_i(z_v))$ and calculate the corresponding loss $\mathcal{L}_i^{\text{pseudo}} = \text{CE}(\mathcal{H}_i(z_v), \hat{y}_i)$ and its gradient $\nabla \mathcal{L}_i^{\text{pseudo}}$. However, if $\mathcal{H}_i$ makes a wrong prediction, $\nabla \mathcal{L}_i^{\text{pseudo}}$ will deviate from the true gradient. To optimize the accuracy of $\nabla \mathcal{L}_i^{\text{pseudo}}$, we refine pseudo-labels by evaluating the consistency between the model's current prediction and its historical behavior. This is achieved by maintaining an Exponential Moving Average (EMA) of past predicted distributions and using the Kullback-Leibler (KL) Divergence to assess consistency. For each client $i$, we maintain an EMA of predicted distributions $\bar{\mathbf{E}}[\tilde{y}_i]$ up to timestep $t-1$, updated as:

$$\bar{\mathbf{E}}[\tilde{y}_i]^{(t)} = \gamma \cdot \bar{\mathbf{E}}[\tilde{y}_i]^{(t-1)} + (1 - \gamma) \cdot \tilde{y}_i^{(t)} \tag{8}$$

where $\gamma$ is a decay rate controlling the weight of the new prediction $\tilde{y}_i^{(t)}$. A smaller $\gamma$ focuses more on historical information, resulting in a smoother estimate of the model's expected output. To evaluate prediction reliability, we compute the KL Divergence between the current prediction $\tilde{y}_i^{(t)}$ and the historical EMA $\bar{\mathbf{E}}[\tilde{y}_i]^{(t-1)}$:

$$D_{\text{KL}} \left( \tilde{y}_i^{(t)} \| \bar{\mathbf{E}}[\tilde{y}_i]^{(t-1)} \right) = \sum_{k=1}^{K} \tilde{y}_{i,k}^{(t)} \cdot \log \left( \frac{\tilde{y}_{i,k}^{(t)}}{\bar{\mathbf{E}}[\tilde{y}_i]_k^{(t-1)}} \right) \tag{9}$$

A small KL value indicates that the current prediction aligns well with the historical behavior, suggesting that the model is confident in its output. Conversely, a large KL value signifies a significant deviation from the historical pattern, which may indicate potential unreliability in the prediction. To select the pseudo-label, we compare the KL Divergence to a threshold $\theta$. If $D_{\text{KL}} < \theta$, we use the current prediction's argmax as the label. Otherwise, we use the argmax from the historical EMA:

$$\hat{y}_i^{(t)} = \begin{cases} \arg \max_k \tilde{y}_{i,k}^{(t)} & \text{if } D_{\text{KL}} \left( \tilde{y}_i^{(t)} \| \bar{\mathbf{E}}[\tilde{y}_i]^{(t-1)} \right) < \theta, \\ \arg \max_k \bar{\mathbf{E}}[\tilde{y}_i]_k^{(t-1)} & \text{if } D_{\text{KL}} \left( \tilde{y}_i^{(t)} \| \bar{\mathbf{E}}[\tilde{y}_i]^{(t-1)} \right) \geq \theta. \end{cases} \tag{10}$$

This optimization ensures that pseudo-labels are reliable by using the current prediction when consistent, and historical predictions when deviations are detected.

### 3.5 STEALTH ENHANCEMENT

In a federated system such as FedSage+, standard server-side defenses (e.g., Krum (Blanchard et al., 2017), trimmed-mean (Yin et al., 2018)) primarily target anomalous gradients produced during the classifier's federated training; however, our poisoning is staged earlier, in the NeighGen training phase, and the adversary acts as a fully benign participant during classifier training. As a consequence, gradients submitted during classifier training are indistinguishable from those of honest clients and evade aggregation-based detectors. To evaluate and improve the covert quality of our attack we therefore focus on the NeighGen stage and measure stealth along three axes: (1) visual and statistical similarity of node distributions after graph mending, and (2) resilience to simple anomaly detectors based on Euclidean distance (Blanchard et al., 2017; Lyu et al., 2024) and cosine similarity (Cao et al., 2020) of update. By evaluating the attack on these axes, we identified concrete weaknesses in its concealment. Guided by these findings, we developed stealth enhancement for IGPA, which reduce structural deviations and align the adversary's update signatures with those of benign clients, while preserving the attack's effectiveness. Further empirical evaluation against a gradient sanitization defense, which applies clipping and noising to shared gradients, is provided in Appendix A.5.

#### 3.5.1 GRAPH SIMILARITY TO BENIGN

Consider a basic defense: because NeighGen acts as a data-augmentation in FedSage+, the victim observes the augmented output (the mended graph). Large deviations between adversarial and clean NeighGen outputs therefore increase detectability. To illustrate this, we visualize the victim's mended graph under clean and attacked conditions in Figure 2 and report the node insertion ratio $\rho$. The results show that, to maximize attack success, adversarial NeighGen must insert an excessively large number of nodes, producing mended graphs that are readily distinguishable from clean ones and thus lack stealth. This behavior arises because the adversarial optimization often follows a markedly different direction from the benign optimization: the adversarial gradient $\nabla\mathcal{L}_j$ can differ substantially from the benign gradient $\nabla\mathcal{L}_i$, which itself is a strong signal of manipulation. To preserve stealth we therefore introduce a scaling factor $\beta \in [0,1]$ that smooths the poisoned update, replacing $\nabla\mathcal{L}_j$ with $\beta\nabla\mathcal{L}_j$. The modified poisoned gradient is expressed as:

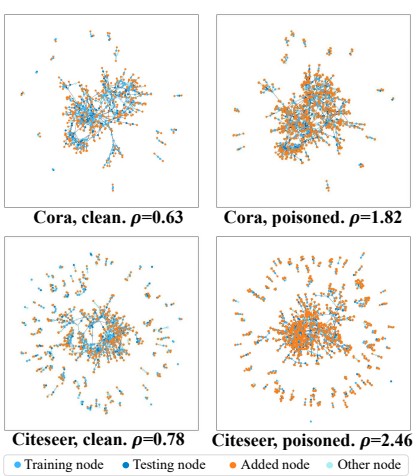

**Cora, clean.** $\rho=0.63$    **Cora, poisoned.** $\rho=1.82$

**Citeseer, clean.** $\rho=0.78$    **Citeseer, poisoned.** $\rho=2.46$

● Training node   ● Testing node   ● Added node   ● Other node

Figure 2: Mended subgraph of a random victim. $\rho$: node insertion ratio.

$$\nabla\mathcal{L}_j = \frac{\beta}{\alpha}\left(\frac{\mathcal{H}_i - \mathcal{H}_{\text{adv}}}{\eta} - \nabla\mathcal{L}_i\right) - \beta\sum_{m\in[M]/\{i,j\}}\nabla\mathcal{L}_m, \qquad (11)$$

By shrinking the adversarial gradient toward the benign gradient, $\beta$ reduces the discrepancy between attacked and clean mended graphs, trading off raw attack strength for increased stealth.

#### 3.5.2 GRADIENT SIMILARITY FOR STEALTH

We next consider a stronger dynamic defense. Since the victim client receives gradients from all other clients in each round, it can effectively act as a central server. To guard against malicious updates, the server can apply anomaly detection based on cosine similarity or Euclidean distance. Concretely, the victim compares each received gradient with its own local gradient and discards those identified as outliers. Our experiments reveal a clear weakness of the attack: for any benign client $i$, the gradient relationship with the malicious client $j$ differs sharply from that with another benign client $m \in$

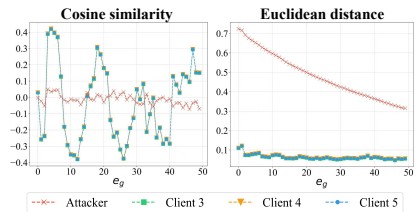

Figure 3: Cosine similarity/Euclidean distance with victim's local gradient

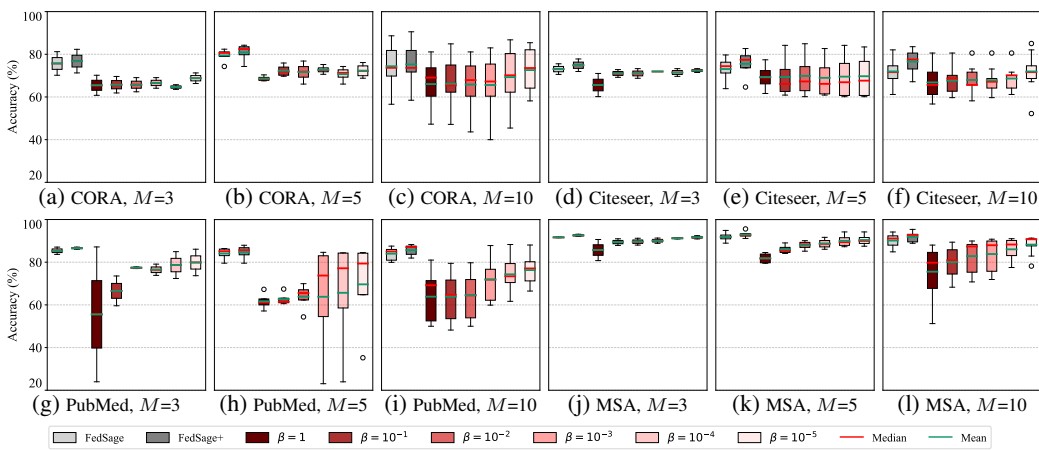

Figure 4: Test accuracy distribution across clients for different datasets and $M$.

$[M] \setminus \{i, j\}$. This pattern holds across both metrics; for example, $|\cos(\nabla \mathcal{H}_i, \nabla \mathcal{H}_m)| \gg |\cos(\nabla \mathcal{H}_i, \nabla \mathcal{H}_j)|$, while $\|\nabla \mathcal{H}_i - \nabla \mathcal{H}_m\| \ll \|\nabla \mathcal{H}_i - \nabla \mathcal{H}_j\|$. Such disparities indicate that the attack is insufficiently stealthy and can be detected during training.

To make the attack more covert, we introduce an adaptive loss to balance the attack objectives while ensuring that the gradient of the adversarial NeighGen $\mathcal{H}_{\text{adv}}$ remains similar to those of benign clients, thereby avoiding detection through gradient-based anomaly detection. The key modification to the training process involves incorporating an additional loss term, $\mathcal{L}^{\mathcal{H}}$, which ensures that the gradient of $\mathcal{H}_{\text{adv}}$ is consistent with those of benign clients. The overall objective function is formulated as a composite loss function:

$$\mathcal{L}_{\text{adv}} = \frac{\mathcal{L}^{\mathcal{H}}}{\bar{\mathbf{E}}[\mathcal{L}^{\mathcal{H}}] + \epsilon} - \lambda \cdot \frac{\mathcal{L}_{\text{mis}}}{\bar{\mathbf{E}}[\mathcal{L}_{\text{mis}}] + \epsilon} \tag{12}$$

Here, the terms are normalized by their respective running expectations, $\bar{\mathbf{E}}[\cdot]$, to maintain scale invariance during optimization. The weighting between the two objectives is dynamically adjusted using the adaptive parameter $\lambda$, which is computed as:

$$\lambda = \sigma \left( \frac{\bar{\mathbf{E}}[\mathcal{L}_{\text{mis}}]}{\bar{\mathbf{E}}[\mathcal{L}^{\mathcal{H}}] + \epsilon} \right) \tag{13}$$

The sigmoid function $\sigma(\cdot)$ ensures that $\lambda$ remains within the range $(0, 1)$, balancing the attack's effectiveness and the need for stealth. By dynamically adjusting $\lambda$ during training, the attack can shift focus between maintaining gradient similarity and maximizing misclassification, depending on the difficulty of each objective at any given step. To further stabilize the training process, the expected values of the loss terms are updated using EMA, as defined in Eq. 8:

$$\bar{\mathbf{E}}[\mathcal{L}_*]^{(t)} = \gamma \cdot \bar{\mathbf{E}}[\mathcal{L}_*]^{(t-1)} + (1 - \gamma) \cdot \mathcal{L}_*^{(t)} \tag{14}$$

where $\mathcal{L}_*$ represents either $\mathcal{L}^{\mathcal{H}}$ or $\mathcal{L}_{\text{mis}}$. This enables the adversarial NeighGen to adapt its behavior over time, ensuring that IGPA remains both potent and undetectable by gradient-based defenses.

## 4 EXPERIMENTS

### 4.1 DATASETS AND EXPERIMENT SETUP

Following FedSage (Zhang et al., 2021), we evaluate on four standard networks: Cora, Citeseer, PubMed, and MSAcademic. They are partitioned into 3, 5, or 10 non-overlapping subgraphs of roughly equal size using the Louvain algorithm. Each partition serves as a client, and we randomly designate one client as the attacker while others are victims; performance is reported on victims only. Full dataset statistics and partition details are provided in Appendix A.2.

We follow the setting in the original FedSage+, including a two-layer GraphSAGE model using the mean aggregator, with 5 neighbors sampled per layer. The model is trained with a batch size of 64

Table 1: Benign clients' test accuracy across different datasets

| Method | CORA | | | Citeseer | | | PubMed | | | MSAcademic | | |
|---|---|---|---|---|---|---|---|---|---|---|---|---|
| | M=3 | M=5 | M=10 | M=3 | M=5 | M=10 | M=3 | M=5 | M=10 | M=3 | M=5 | M=10 |
| FedSage+ | 76.80 | 80.89 | 75.25 | 74.71 | 75.57 | 76.62 | 86.57 | 84.69 | 85.64 | 92.60 | 92.77 | 91.05 |
| $\beta$=1 | 64.65 | 68.67 | 65.62 | 65.61 | 68.99 | 66.83 | 55.56 | 61.68 | 63.85 | 85.72 | 82.55 | 75.58 |
| $\beta$=$10^{-1}$ | 65.47 | 70.74 | 65.83 | 71.04 | 69.36 | 67.66 | 66.57 | 62.73 | 63.87 | 89.45 | 86.06 | 77.83 |
| $\beta$=$10^{-2}$ | 65.75 | 71.67 | 66.05 | 71.04 | 69.36 | 67.83 | 76.46 | 63.81 | 64.53 | 89.65 | 87.98 | 81.47 |
| $\beta$=$10^{-3}$ | 65.75 | 72.35 | 66.24 | 71.49 | 69.55 | 67.99 | 77.41 | 63.86 | 71.87 | 90.10 | 88.75 | 83.35 |
| $\beta$=$10^{-4}$ | 66.58 | 72.35 | 69.32 | 71.95 | 69.74 | 68.66 | 78.66 | 65.68 | 74.33 | 91.25 | 90.05 | 84.95 |
| $\beta$=$10^{-5}$ | 68.79 | 72.81 | 72.61 | 72.40 | 69.93 | 71.97 | 79.88 | 69.64 | 76.32 | 91.70 | 90.43 | 87.16 |

for 300 epoches, using the Adam optimizer with a learning rate of 0.001. The data is divided into 60% training, 20% validation, and 20% testing. The hyperparameter $\alpha$ is set to 1. The total number of epochs for the joint training of NeighGen, $e_g$, is a predetermined system hyperparameter. Our attack is both rapid and robust, effective even with a limited budget of $e_g = 20$ and maintaining it across a wide range up to $e_g = 100$. Nonetheless, to ensure a strict and conservative evaluation, we set $e_g = 50$ for all main experiments, as it constituted a particularly challenging scenario for the attacker under minimal attack intensity in our robustness analysis (see Appendix A.3).

We conduct experiments on FedSage+ to get clean baselines for the attack. For attack evaluation, we vary the attack intensity parameter $\beta \in \{1, 10^{-1}, 10^{-2}, 10^{-3}, 10^{-4}, 10^{-5}\}$. As our primary evaluation metric, we use node classification accuracy computed on queries sampled from the testing nodes of each local subgraph. All results are averaged over five independent runs with different random seeds. All experiments were conducted on a server equipped with 4 NVIDIA GeForce RTX 3090 GPUs, 64GB RAM, and Python 3.8.

## 4.2 OVERALL PERFORMANCE

### 4.2.1 ATTACK EFFECTIVENESS

**The proposed IPGA consistently degrades classification performance by over 10% across all datasets and client sizes when $\beta = 1$, with the impact diminishing as $\beta$ decreases.** We report the test accuracy of benign clients under different attack intensities in Table 1. The results demonstrate that even with minimal attack intensities (e.g., $\beta = 10^{-5}$), the final model's accuracy is consistently lower than that of the clean FedSage+, confirming the attack's effectiveness. For example, the accuracy drops from 80.89% to 72.81% on CORA ($M$=5) and from 86.57% to 79.88% on PubMed ($M$=3). Moreover, the attack's effectiveness is inversely correlated with the intensity parameter $\beta$. As $\beta$ increases from $10^{-5}$ to 1, the model's accuracy consistently declines in most settings, illustrating that the attacker can flexibly control the severity of the disruption. For instance, on dataset MSAcademic ($M$=10), the accuracy drops from 87.16% to 75.58% as $\beta$ increases.

Additionally, Figure 4 presents the distribution of attack impacts across individual clients. The boxplot results indicate that the local test accuracy distributions of benign clients shift downward and exhibit increased variability due to the attack, suggesting that the performance degradation is global and not limited to specific clients.

### 4.2.2 ATTACK STEALTHINESS

**IPGA demonstrates stealthiness across multiple dimensions, including graph structure similarity, gradient indistinguishability, and loss convergence behavior.** First, we assess the stealthiness of the mended graph structure as discussed in Section 3.5.1. In Figure 5, we visualize the mended graphs under different attack intensities, including clean FedSage+ and attacks with $\beta = 1$, $10^{-2}$, $10^{-4}$, and $10^{-5}$. As shown, when $\beta = 1$ or $10^{-2}$, the attack is strong, and NeighGen inserts a substantial number of new nodes to maximize disruption of the original graph, resulting in poor stealthiness. In contrast, when $\beta = 10^{-4}$ or $10^{-5}$, fewer nodes are inserted by the adversarial Neigh-Gen, and the graph becomes increasingly similar to the clean FedSage+ graph. We quantitatively report the node insertion ratio in Figure 5, where the results show that, at lower attack intensities, the number of inserted nodes closely resembles that of the clean FedSage+ case.

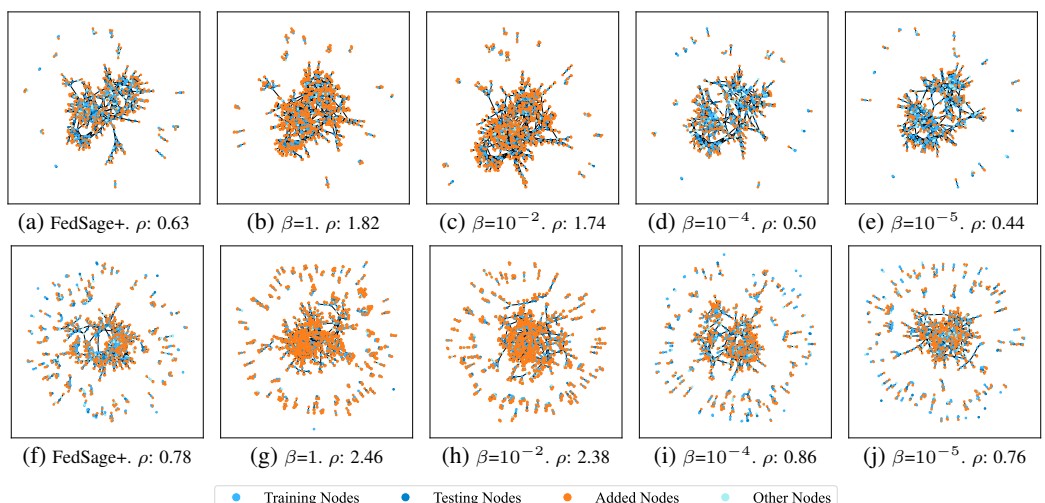

(a) FedSage+. $\rho$: 0.63    (b) $\beta$=1. $\rho$: 1.82    (c) $\beta$=$10^{-2}$. $\rho$: 1.74    (d) $\beta$=$10^{-4}$. $\rho$: 0.50    (e) $\beta$=$10^{-5}$. $\rho$: 0.44

(f) FedSage+. $\rho$: 0.78    (g) $\beta$=1. $\rho$: 2.46    (h) $\beta$=$10^{-2}$. $\rho$: 2.38    (i) $\beta$=$10^{-4}$. $\rho$: 0.86    (j) $\beta$=$10^{-5}$. $\rho$: 0.76

Training Nodes    Testing Nodes    Added Nodes    Other Nodes

Figure 5: Visualization of mended subgraph on a random victim. (a)-(e): a victim on Cora. (f)-(j): a victim on Citeseer. $\rho$: node insertion ratio.

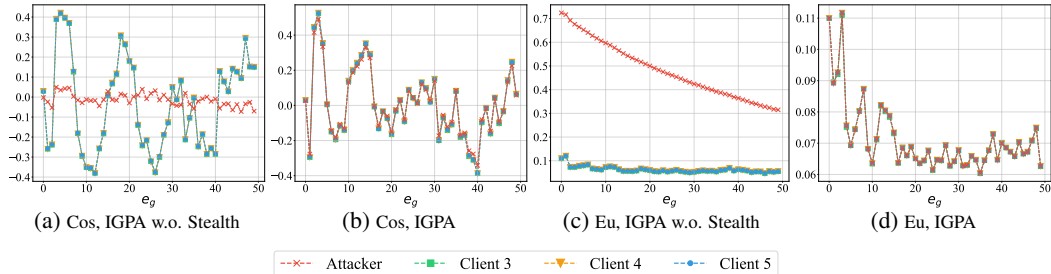

(a) Cos, IGPA w.o. Stealth    (b) Cos, IGPA    (c) Eu, IGPA w.o. Stealth    (d) Eu, IGPA

Attacker    Client 3    Client 4    Client 5

Figure 6: **Cos**ine similarity/**Eu**clidean distance with a victim's local gradient

Second, we analyze the stealthiness in gradient transmission. As shown in Figure 6, after the enhancement in Section 3.5.2, the gradients sent by the attacker to a victim client are sufficiently similar to those from benign clients, making them difficult to distinguish. Notably, we observe that gradients among benign clients exhibit remarkably high similarity, with numerical differences below the order of $10^{-4}$, which makes concealing the attack even more challenging(see Appendix A.4.1 for details). We further provide a detailed analysis of how each layer in the NeighGen architecture impacts gradient similarity (see Appendix A.4.1). This indicates that our attack can evade anomaly-based detection defenses that rely on metrics like cosine similarity or Euclidean distance.

Finally, the convergence behavior of the NeighGen training loss on the victim clients provides another indicator of stealthiness (see Appendix A.4.2 for details). Besides, the accuracy curves of GraphSage during federated training further illustrate the covert nature of the attack, as discussed in Appendix A.4.3.

## 5 CONCLUSION

In this work, our study reveals a critical vulnerability in auxiliary-information–sharing frameworks for federated graph learning, exemplified by FedSage+. We introduce IGPA, an indirect data-manipulation attack that poisons shared generator gradients so that benign clients themselves generate corrupted nodes, sharply degrading node-classification accuracy across datasets and client scales. The attack remains effective even under strong defenses and is difficult to detect, as its gradients and synthetic nodes closely mimic benign behavior. These findings show that relaxing information boundaries, while improving utility, dramatically enlarges the attack surface, exposing systems not only to privacy leakage but also to stealthy active compromise. Robust defenses that specifically address collaborative generator sharing are therefore essential for secure, real-world FGL deployment.

## 6 REPRODUCIBILITY STATEMENT

The experiments and results presented in this paper are fully reproducible. The source code used for training and evaluation, along with the configurations for each experiment, is provided as supplementary materials through the conference submission system. For further details on model architecture, hyperparameters, and training procedures, please refer to the main text, specifically Section 4.1.

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

# A APPENDIX

## A.1 LLM USAGE

In this work, large language models (LLMs) were employed as an aid to improve and polish the writing throughout the paper. Specifically, LLMs were used to assist in drafting, refining, and enhancing clarity, coherence, and stylistic aspects of the text. However, the core research ideas, experiments, and results were independently developed and conducted by the authors. The LLM played a supportive role in writing and not in research ideation or the experimental process.

## A.2 DATASETS STATISTICS

We adopt four widely used benchmark graphs: Cora and Citeseer (Sen et al., 2008), PubMed (Namata et al., 2012), and MSAcademic (Shchur et al., 2018). To simulate a distributed subgraph setting, we apply the Louvain community-detection algorithm (Blondel et al., 2008) for hierarchical clustering on each dataset. For each run, the graph is partitioned into 3, 5, or 10 clusters of roughly equal size, and each cluster is treated as a distinct client. In every partitioning scenario, one client is randomly selected as the attacker while all others act as victims. Attack performance is measured exclusively on victim clients. Table 2 reports the number of nodes, edges, classes, and feature dimensions for each dataset.

Table 2: Statistics of the datasets and the synthesized distributed subgraphs with $M = 3, 5$, and 10.

| Data | Cora | Citeseer | PubMed | MSAcademic |
|---|---|---|---|---|
| #Classes | 7 | 6 | 3 | 15 |
| #Nodes | 2708 | 3312 | 19717 | 18333 |
| #Edges | 5429 | 4715 | 44338 | 81894 |
| $M = 3$ | | | | |
| Avg. #nodes | 903 | 1104 | 6572 | 6111 |
| Avg. #edges | 1675 | 1518 | 12932 | 23584 |
| Missing #edges | 403 | 161 | 5543 | 11141 |
| $M = 5$ | | | | |
| Avg. #nodes | 542 | 662 | 3943 | 3667 |
| Avg. #edges | 968 | 902 | 7630 | 13949 |
| Missing #edges | 589 | 206 | 6189 | 12151 |
| $M = 10$ | | | | |
| Avg. #nodes | 271 | 331 | 1972 | 1833 |
| Avg. #edges | 450 | 442 | 3789 | 5915 |
| Missing #edges | 929 | 300 | 6445 | 22743 |

## A.3 ROBUSTNESS ANALYSIS ON NEIGHGEN TRAINING EPOCHS

The hyperparameter $e_g$, denoting the total number of epochs for the joint training of NeighGen, is predetermined by federated system, placing it beyond the attacker's control. As our adversarial poisoning is executed by propagating malicious gradients during this training phase, the value of $e_g$ directly influences the attack's cumulative effect. To rigorously evaluate the robustness of our attack under this fundamental system parameter and to establish a conservative baseline for the main experiments, we investigate the impact of varying $e_g$ on the attack performance.

We conducted experiments on the Cora and Citeseer with $M = 5$ clients, analyzing the final global classifier's test accuracy under different attack intensities $\beta$ across a range of $e_g$ values from 20 to 100. As illustrated in Figure 7, IGPA consistently maintains stable and effective poisoning across all $e_g$ settings. A key strength of our approach is its efficiency; it achieves a significant degradation in model accuracy even under a very limited training budget (e.g., $e_g = 20$). This indicates that IGPA is rapid. Furthermore, as $e_g$ increases, the attack effectiveness remains robust and is not suppressed by the aggregated gradients from the victim or other benign clients.

Note that the attack effectiveness, measured by the final accuracy drop, does not monotonically improve with larger $e_g$ but exhibits some fluctuation. This behavior is expected, as the victim NeighGen is shaped not only by the malicious gradients but also through the aggregation of benign updates from all benign clients in each round. Notably, the attack effect was relatively weakest for our method at $e_g = 50$ under the minimal attack intensity ($\beta = 10^{-5}$). To ensure a strict and challenging evaluation scenario, we therefore select $e_g = 50$ as the fixed parameter for all main experiments, thereby presenting our results under the most unfavorable conditions for the attacker.

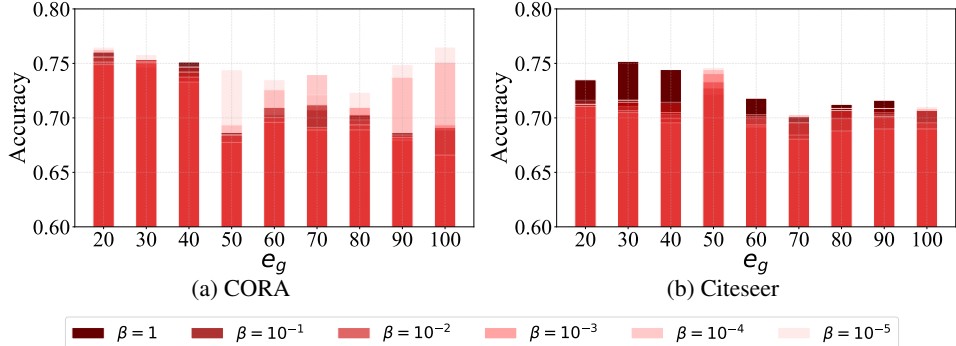

(a) CORA  (b) Citeseer

$\beta = 1$  $\beta = 10^{-1}$  $\beta = 10^{-2}$  $\beta = 10^{-3}$  $\beta = 10^{-4}$  $\beta = 10^{-5}$

Figure 7: clients' test accuracy vs different $e_g$

## A.4 ADDITIONAL STEALTHINESS ANALYSIS

### A.4.1 GRADIENT TRANSMISSION

In section 4.2.2, we analyze the stealthiness in gradient transmission. Here, we further provide a detailed analysis of how each layer in the NeighGen architecture impacts gradient similarity. The NeighGen architecture consists of two fully connected layers and one flatten layer. Using cosine similarity as a detection metric (Figure 8), the primary effect of the stealth enhancement is on the two fully connected layers, while the impact on the flatten layer is limited. This is because the two fully connected layers have relatively fewer parameters ($128 \times 128$ and $128 \times 256$, respectively), whereas the flatten layer has a significantly larger number of parameters ($256 \times 4096$). Consequently, the gradient dimensionality of the flatten layer is much higher, and cosine similarity retains a better discriminative ability in high-dimensional spaces. In contrast, Euclidean distance-based measures fail in high-dimensional spaces. Therefore, IGPA demonstrates strong concealment across all layers, as shown in Figure 9.

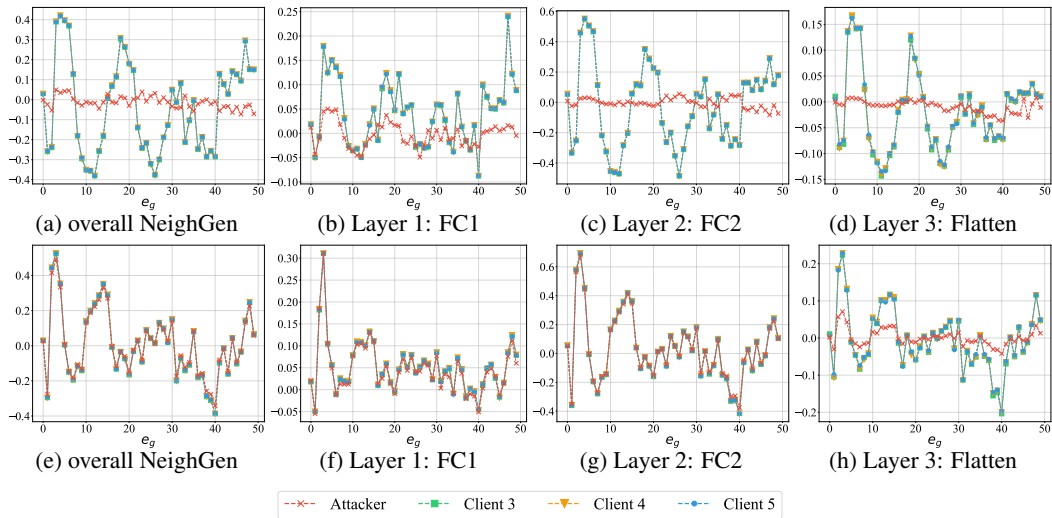

(a) overall NeighGen  (b) Layer 1: FC1  (c) Layer 2: FC2  (d) Layer 3: Flatten

(e) overall NeighGen  (f) Layer 1: FC1  (g) Layer 2: FC2  (h) Layer 3: Flatten

Attacker  Client 3  Client 4  Client 5

Figure 8: **Cosine** similarity with a victim's local gradient. (a)-(d): IGPA w.o. Stealth (e)-(h): IGPA

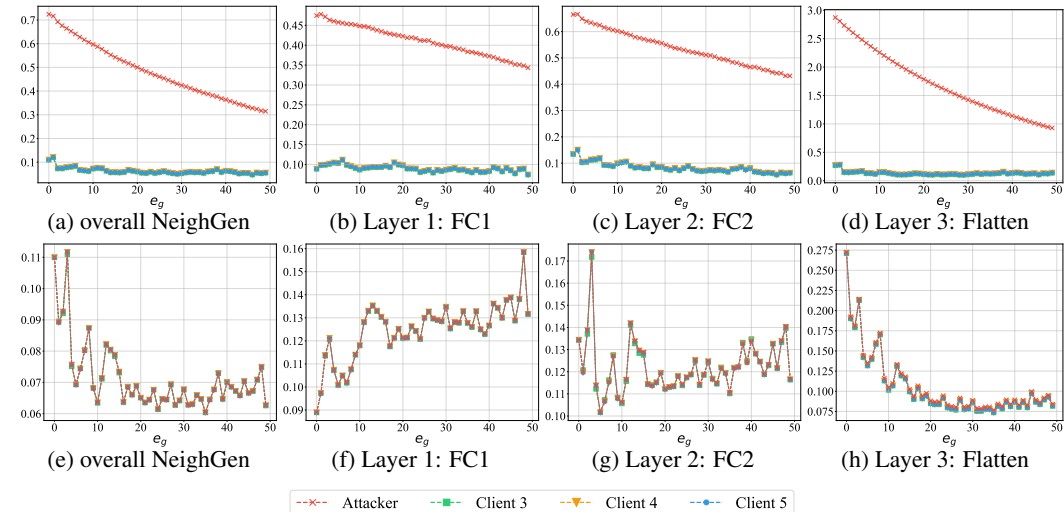

Figure 9: **Euclidean distance** with a victim's local gradient. (a)-(d): IGPA w.o. Stealth (e)-(h): IGPA

Figures 8 and 9 also show that the gradient $\nabla H_j$ received by a victim client $C_i$ from a benign client $C_j$ ($j \in \{3, 4, 5\}$) exhibits remarkably high similarity, with numerical differences below the order of $10^{-4}$ actually, which makes concealing the attack even more challenging. To explain this phenomenon, we analyzed the distribution similarity (based on Kolmogorov-Smirnov testing) and feature similarity (based on correlation coefficients) between the subgraphs of each client. Figure 10 shows that these subgraphs are highly similar in both distribution and features. This explains why, in Figures 8 and 9, benign clients 3, 4, and 5 are so close to one another.

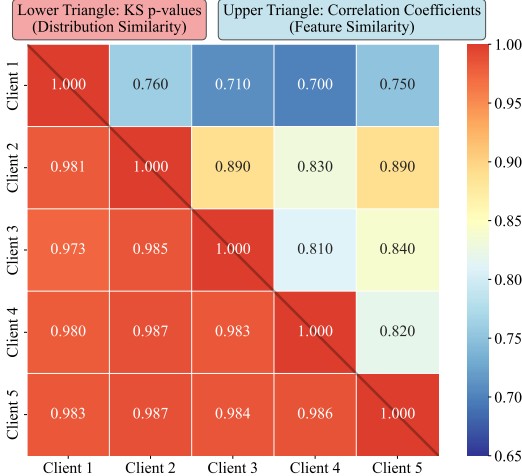

Figure 10: Similarity analysis

### A.4.2 NEIGHGEN TRAINING LOSS

We recorded the NeighGen training loss for each victim client on the Cora dataset with $M = 5$ in Figure 11. As the attack intensity $\beta$ decreases from 1 to $10^{-5}$, the convergence trend of the loss gradually approaches that of the clean FedSage+ scenario. This behavior provides an additional guarantee of the attack's stealthiness.

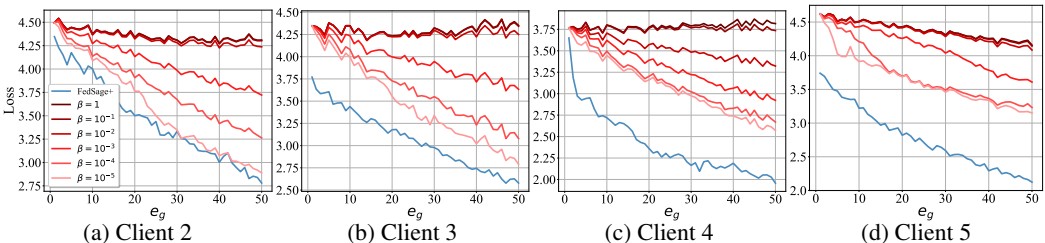

Figure 11: NeighGen's loss v.s. epoch on Cora, M=5

### A.4.3 FEDERATED TRAINING ACCURACY CURVES

We further illustrate the stealthiness through the accuracy curves of GraphSage during federated training. One main advantage of our attack's stealthiness is that we naturally leverage NeighGen, a component already present in FedSage+, without introducing any additional obvious parts. Essentially, our attack is a form of data poisoning. In Figure 12, we report the training accuracy curves for Cora and Citeseer across different values of $M = 3$, 5, and 10, under both clean conditions and various attack intensities. The accuracy curves for the attacked models show a similar growth trend to the clean model, but with a slower rate of increase and a lower ceiling. This illustrates a key characteristic of our attack: it effectively limits the upper bound of FedSage+'s performance through the joint training of NeighGen. This subtle limitation, achieved without any explicit disturbance, is a key stealthiness advantage of our approach.

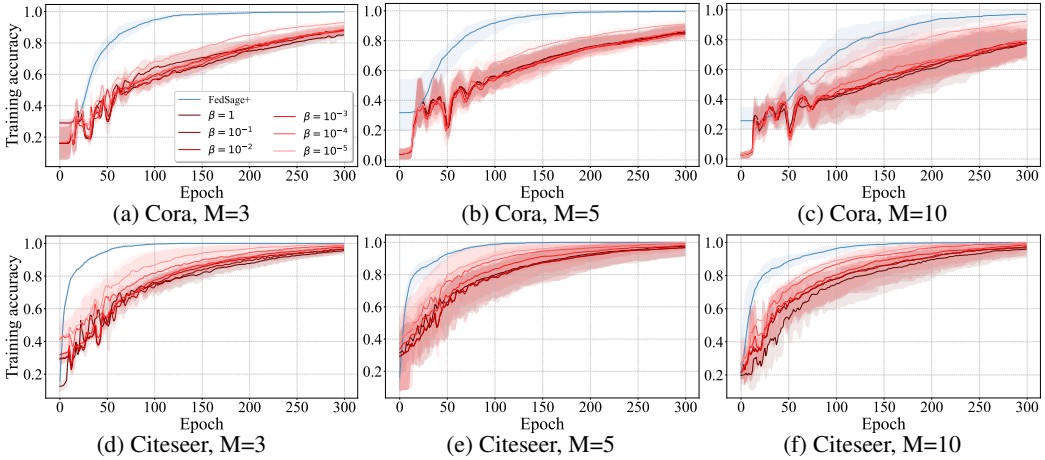

Figure 12: Clients' training accuracy v.s. epoch

## A.5 EVALUATION AGAINST GRADIENT SANITIZATION DEFENSE

To evaluate the robustness of IGPA under a more secure setting, we implement a **Gradient Sanitization** (GS) defense at the server side. This defense directly processes the shared NeighGen gradients before sending out to mitigate potential poisoning. The defense consists of two consecutive operations applied to each client's NeighGen gradient $\nabla \mathcal{L}_i$:

1. **Clipping**: $\nabla \mathcal{L}_i \leftarrow \text{clip}(\nabla \mathcal{L}_i, c)$, which bounds the gradient values to the range $[-c, c]$.

2. **Noising**: $\nabla \mathcal{L}_i \leftarrow \nabla \mathcal{L}_i + \mathcal{N}(0, \sigma^2)$, where $\mathcal{N}(0, \sigma^2)$ is Gaussian noise with variance $\sigma^2 = (0.01 \cdot c)^2$.

The hyperparameter $c$ controls the clipping threshold and directly influences the defense strength. We set $c = 0.01$ based on preliminary validation, as this value balances defense effectiveness against minimal interference with legitimate gradients. This process aims to reduce the magnitude of anomalous gradient components while adding noise to obscure malicious patterns.

All experiments in this section were conducted with a fixed total number of clients $M = 5$ across all datasets. Table 3 reports the global model test accuracy under IGPA, both with and without the gradient sanitization defense applied, for varying attack intensity $\beta$.

Table 3: Global model accuracy (%) under IGPA with the Gradient Sanitization defense across datasets.

| $\beta$ | Cora | | Citeseer | | PubMed | | MSAcademic | |
|---|---|---|---|---|---|---|---|---|
| | w/ GS | w/o GS | w/ GS | w/o GS | w/ GS | w/o GS | w/ GS | w/o GS |
| 1 | 75.15 | 68.67 | 73.35 | 68.99 | 72.82 | 61.68 | 87.91 | 82.55 |
| $10^{-1}$ | 75.28 | 70.74 | 73.68 | 69.36 | 73.45 | 62.73 | 89.27 | 86.06 |
| $10^{-2}$ | 76.52 | 71.67 | 73.14 | 69.36 | 75.83 | 63.81 | 90.15 | 87.98 |
| $10^{-3}$ | 77.09 | 72.35 | 74.23 | 69.55 | 75.42 | 63.86 | 90.86 | 88.75 |
| $10^{-4}$ | 77.31 | 72.35 | 74.71 | 69.74 | 78.68 | 65.68 | 91.42 | 90.05 |
| $10^{-5}$ | 77.64 | 72.81 | 74.89 | 69.93 | 78.15 | 69.64 | 91.85 | 90.43 |

The gradient sanitization defense provides partial mitigation. Across all datasets and $\beta$ values, the accuracy with defense is consistently higher than without defense. However, the defense cannot fully neutralize IGPA. The performance degradation persists compared to the non-attacked baseline. The defense effectiveness is **contingent on attack stealthiness**. When IGPA generates more subtle poisoned gradients (by reducing perturbation magnitude), the accuracy difference between defended and undefended scenarios diminishes. This occurs because the sanitization operations have limited discriminative power against gradients that already fall within the legitimate range.

