# OpenReview forum: "Sharing is Sabotaging: Cross-Client Poisoning Attacks on Federated Graph Learning"
_ICLR.cc/2026/Conference — Submitted to ICLR 2026_

### Official Review · Reviewer_vVHC · 2025-10-27

**Soundness:** 2
**Presentation:** 3
**Contribution:** 2
**Rating:** 2
**Confidence:** 3

**Summary:**

The paper introduces a new poisoning attack, Indirect Gradient Poisoning Attack (IGPA), targeting auxiliary-information–sharing frameworks in Federated Graph Learning (FGL), such as FedSage+. Unlike traditional attacks that directly tamper with model parameters or local data, IGPA manipulates the gradients exchanged during the training of neighbor generators (NeighGen). By injecting malicious updates into these shared gradients, an adversarial client can cause benign clients to generate poisoned synthetic nodes, which subsequently degrade the global model’s performance. The attack operates in a gray-box setting, maintains stealth by aligning adversarial gradients with benign ones, and remains effective across datasets (Cora, Citeseer, PubMed, MSA).

**Strengths:**

- This paper introduces an innovative poisoning vector exploiting gradient sharing in generator-based FGL, expanding the attack surface beyond known methods.
- This paper provides a rigorous problem formulation and detailed methodological breakdown (surrogate model, gradient crafting, stealth enhancement.

**Weaknesses:**

- The main concern is the weak attack performance reported in Table 1. The maximum attack performance does not exceed 50% for all four datasets. The reviewer is not sure whether such performance is convincing. In addition, as discussed in Section 3.5, the stealthiness seems to be highly sensitive to \belta and \lambda. Given the limited discussion on these two key parameters, the reviewer may be concerned about the effectiveness of the proposed attack.
- The proposed scheme seems to have a limited scope with FedSage+. If there is no available node generator in the framework, the proposed attack may lose its functionality.

**Questions:**

- What is the sensitivity of \lambda?
- It suggests providing detailed defense results rather than similarity checks.
- Since the reported attack performance is not that impressive, the reviewer may think a backdoor attack would succeed in such a scenario. That is to say, the system may keep a high overall performance while losing accuracy on specific tasks. It suggests further exploration.

---

> ### Author Response · Authors · 2025-11-29
> **Response to Reviewer vVHC**
>
> ## Weakness 1: Attack Performance
>
> Please refer to General Response To All Reviewers - Attack Performance.
>
> ## Weakness 2: Generalizability of IGPA
>
> Please refer to General Response To All Reviewers - Generalizability of IGPA.
>
> ## Question 1: Sensitivity of λ
>
> Thank you for your question regarding the sensitivity of λ. We would like to clarify that **λ is not a fixed hyperparameter** that requires manual tuning, but rather an **automatically adapted parameter** that dynamically adjusts during training.
>
> As defined in Equation (13) in our manuscript:
>
> $$\lambda = \sigma\left( \frac{\mathbf{\bar{E}}[\mathcal{L} _ {\text{mis}}]}{\mathbf{\bar{E}}[\mathcal{L}^{\mathcal{H}}] + \epsilon} \right)$$
>
> This design ensures that λ **automatically balances** the trade-off between attack effectiveness ($\mathcal{L}_{\text{mis}}$) and stealthiness ($\mathcal{L}^{\mathcal{H}}$) based on real-time training dynamics. When misclassification becomes easier to achieve, λ increases to focus more on stealth; conversely, when maintaining gradient similarity is challenging, λ decreases to preserve attack potency.
>
> Through extensive experiments across different datasets and client configurations, we found this adaptive mechanism to be **highly robust and insensitive** to initial conditions. The EMA smoothing provides stable evolution of λ throughout training, eliminating the need for sensitive hyperparameter tuning.
>
> ## Question 2: Detailed defense results
>
> Thank you for this valuable suggestion regarding defense evaluation. We agree that demonstrating the attack's effectiveness against actual defense mechanisms, rather than just similarity checks, would provide stronger validation.
>
> In our revised manuscript, we have expanded the defense evaluation to include results against state-of-the-art defense suggested by Reviewer 726Z. Please refer to General Response To All Reviewers - Stronger defense.
>
> ## Question 3: Backdoor attack exploration
>
> Thank you for this insightful observation regarding backdoor attacks. We appreciate the opportunity to clarify the relationship between IGPA and potential backdoor attacks in FGL.
>
> The successful demonstration of IGPA indeed establishes the feasibility of using generator poisoning as an attack vector in FGL systems. While IGPA aims to **degrade overall model performance while remaining undetectable**, the backdoor attack scenario you mentioned would pursue a different objective: **degrading accuracy on specific tasks or classes while maintaining high overall performance**.
>
> This distinction highlights two important aspects:
>
> 1. Attack Vector Validation: IGPA demonstrates that the generator-sharing mechanism in FGL creates a viable pathway for model manipulation, which could potentially be adapted for backdoor attacks.
>
> 2. Different Attack Objectives:
>    - IGPA: Global performance degradation
>    - Backdoor attack: Targeted performance degradation
>
> We agree that exploring backdoor attacks in this context represents a promising and important direction for future work. The generator poisoning framework established in our work could be extended to study how malicious clients might embed targeted backdoors through carefully crafted gradient manipulations.

---

### Official Review · Reviewer_t3Eo · 2025-10-27

**Soundness:** 3
**Presentation:** 2
**Contribution:** 2
**Rating:** 4
**Confidence:** 3

**Summary:**

The manuscript introduces a novel gray-box poisoning attack in FGL that exploits shared neighbor representations during training. By targeting auxiliary-information-sharing frameworks, a malicious client can inject compromised gradients during generator updates. This manipulation results in benign clients incorporating poisoned synthetic nodes into their local subgraphs, leading to persistent model degradation as corrupted signals propagate during training. The authors formalize this threat model and propose a stealthy, optimization-based attack that is model-agnostic and effective across varying datasets, partition strategies, and client configurations. Their findings expose a critical vulnerability in the data-generation layer of FGL systems and underscore the need for robust defense mechanisms.

**Strengths:**

The manuscript presents a compelling and technically rigorous contribution to the field of FGL by identifying a novel and underexplored security threat. The proposed IGPA is conceptually innovative, exploiting the shared neighbor-generation mechanism in auxiliary-information-sharing frameworks like FedSage+. Unlike conventional models or data poisoning, IGPA operates indirectly through gradient manipulation of generator updates, making the attack highly stealthy and persistent. This manuscript is methodologically sound, offering clear mathematical formalization, algorithmic design, and theoretical justification for both the attack and its stealth enhancements. The empirical evaluation is comprehensive, covering multiple benchmark datasets and varying numbers of clients. Results consistently demonstrate the attack’s effectiveness in degrading global model performance and its resilience under different configurations and stealth constraints. Visualizations and quantitative analyses of gradient similarity, node insertion ratios, and convergence behavior further strengthen the experimental credibility.

**Weaknesses:**

While the manuscript demonstrates strong methodological rigor and novelty, several limitations remain that could weaken its overall impact and generalizability.

The proposed IGPA is evaluated exclusively on FedSage+, which, although representative of auxiliary-information-sharing frameworks, may not encompass the full diversity of FGL architectures. Other frameworks (e.g., FedGNN, FedGCN, or personalized subgraph FGL) employ different data-sharing or aggregation mechanisms that might not be equally vulnerable. The absence of comparative evaluation across these settings restricts the claim of model-agnostic applicability.

The gray-box assumption, where the adversary has access to shared embeddings and generators, simplifies real-world complexities. In many practical deployments, such access may be limited or obfuscated by encryption, compression, or secure aggregation protocols. This manuscript does not thoroughly discuss how these protective mechanisms would affect the feasibility or strength of IGPA.

This manuscript convincingly exposes a new vulnerability but provides minimal insight into how such attacks could be mitigated. It neither analyzes the limitations of existing defenses in detail nor proposes preliminary countermeasures tailored to generator poisoning. This omission leaves the work somewhat one-sided, emphasizing offensive capability without offering a path toward practical system hardening.

The experiments focus primarily on node classification accuracy as the sole performance indicator. Other important metrics, such as communication efficiency, robustness–utility trade-offs, or computational overhead, are not reported. These would help characterize the full impact of the attack and its practical relevance under resource-constrained federated environments.

The evaluation uses well-known benchmark datasets (Cora, Citeseer, PubMed, MSAcademic) with relatively small graph sizes. The scalability and persistence of IGPA in large-scale or dynamic graph environments (e.g., evolving social or financial networks) remain unexplored. Real-world systems may introduce additional constraints, (e.g., heterogeneous connectivity, asynchronous updates, or variable communication delays) that could affect both attack feasibility and detectability.

While the attack’s formulation is mathematically defined, the manuscript offers little theoretical treatment of how IGPA impacts convergence properties of federated optimization. A more formal examination of gradient divergence, variance amplification, or convergence delay would strengthen the analytical depth of the work.

**Questions:**

Please refer to ``weaknesses`` part.

---

> ### Author Response · Authors · 2025-11-29
> **Response to Reviewer t3Eo**
>
> ## Weakness 1: Generalizability of IGPA
>
> Please refer to General Response To All Reviewers - Generalizability of IGPA.
>
> ## Weakness 2: gray-box assumption
>
> Thank you for the feedback. We appreciate the opportunity to clarify the realism and relevance of our assumptions. Our gray-box threat model is in fact carefully aligned with the operational requirements of auxiliary-information-sharing FGL frameworks like FedSage+:
>
> **Minimal and Necessary Access**: The adversary possesses **exactly the same knowledge and capabilities as any other participating client** in the FGL system. As a legitimate participant in the federated learning process, the malicious client naturally receives shared generators and node embeddings – this is not a special privilege but a basic requirement for participating in the collaborative training framework. The adversary does not have elevated access or additional information beyond what normal operation grants to all clients, which makes our threat model particularly realistic and concerning.
>
> **Reflective of Real FGL Deployment**: The very purpose of auxiliary-information-sharing FGL is to enable performance gains through strategic sharing of graph-related information. In practical deployments, frameworks like FedSage+ **do share generators and embeddings**. This is their core operational mechanism, not a theoretical convenience.
>
> **Compatibility with Protection Mechanisms**: While encryption and secure aggregation are valuable for protecting model parameter updates, they typically **do not apply to the auxiliary information layer**. The generator and embedding sharing occurs at the data-augmentation level, which must remain accessible for the collaborative neighbor generation process to function.
>
> Our threat model therefore captures a realistic and concerning scenario: an adversary operating within the normal bounds of FFL participation, exploiting the very information sharing that enables the system's improved performance. This represents a practical vulnerability in real-world FGL deployments that rely on auxiliary information sharing.
>
> ## Weakness 3: Lack of defense
>
> Please refer to General Response To All Reviewers - Stronger defense.
>
> ## weakness 4: Evaluation metrics other than accuracy
>
> Thank you for this valuable feedback regarding evaluation metrics. We agree that evaluating the full system impact of IGPA is important for understanding its practical implications.
>
> **Communication Overhead:**
> Our attack introduces **zero additional communication overhead**, as the malicious client strictly follows the same communication protocol as all benign participants. The attack operates entirely within the existing gradient and model sharing mechanisms of FedSage+.
>
> **Computational Cost:**
> The adversary incurs approximately **2× the computational cost of a benign client**, due to the simultaneous training of two components: A benign surrogate model and An adversarial NeighGen model.
>
> However, both components can be **pre-trained offline** using only the adversary's local data, without requiring any information from benign clients. This means the additional computation does not impact online training efficiency or delay federated updates.
>
> **Utility–Robustness Trade-off:**
>
> Our results demonstrate a clear trade-off: Stronger attacks (higher β) cause greater accuracy degradation but reduce stealth, while weaker attacks (lower β) maintain better stealth but with reduced effectiveness. This continuum allows adversaries to balance attack impact against detection risk based on their objectives.
>
> ## weakness 5: Dataset scale
>
> Thank you for raising this important point regarding dataset scale and real-world applicability. Our evaluation uses standard FGL benchmark datasets (Cora: 2.7K nodes, Citeseer: 3.3K, PubMed: 19K, MS Academic: 18K) that are relatively modest in scale. More importantly, we would like to emphasize that **the core vulnerability we identify**, gradient poisoning through shared generators, **is architectural and independent of graph size**. Our attack mechanism targets the NeighGen module's collaborative training process, which operates at the client level rather than the global graph scale. Since each client only processes its local subgraph regardless of the overall graph size, the attack feasibility remains consistent as the system scales.

---

> ### Author Response · Authors · 2025-11-29
> **Response to Reviewer t3Eo**
>
> ## weakness 6: Theoretical analysis
>
> Thank you for this insightful comment regarding the analysis of convergence properties. We agree that a formal examination of how IGPA affects federated optimization would strengthen the analytical depth of our work.
>
> In our current manuscript, we have included empirical analyses of these aspects in the appendix:
>
> - **Appendix A.4.2** presents the convergence behavior of NeighGen training loss under different attack intensities
> - **Appendix A.4.3** shows the federated training accuracy curves across epochs
> - **Appendix A.4.1** provides detailed analysis of gradient divergence through similarity metrics
>
> These empirical results consistently demonstrate that IGPA causes significant convergence degradation while maintaining stealthy behavior patterns. Specifically, our experiments show that the attack effectively limits the upper bound of model performance without introducing obvious disturbance to the training dynamics.

---

### Official Review · Reviewer_726Z · 2025-10-30

**Soundness:** 3
**Presentation:** 3
**Contribution:** 3
**Rating:** 8
**Confidence:** 4

**Summary:**

This paper focuses on a critical vulnerability in auxiliary-information-sharing Federated Graph Learning (FGL) frameworks, with FedSage+ as a representative example. It proposes a gray-box poisoning threat model for auxiliary-information-sharing FGL, where a malicious client poisons shared graph-related gradients to compromise the global model—without accessing other clients’ raw graph data or local parameters. Introduces the Indirect Gradient Poisoning Attack (IGPA), a gradient-guided poisoning method that approximates benign update directions and injects malicious signals. This attack reduces node classification accuracy while maintaining high stealth. This paper Empirically demonstrates IGPA’s performance across four standard datasets (Cora, Citeseer, PubMed, MSAcademic) and different client scales (3, 5, 10 clients). It shows the attack degrades model accuracy by over 10% (when β=1) and evades standard defenses through graph structure similarity and consistent loss convergence.

**Strengths:**

(1)The paper addresses an understudied vulnerability in auxiliary-information-sharing FGL frameworks. Unlike prior work focusing on privacy leakage, it targets active, indirect data manipulation via generator gradient poisoning—filling a critical gap in FGL security research.
(2)IGPA’s two-stage mechanism (training a surrogate classifier + optimizing an adversarial NeighGen) is technically sound. The use of pseudo-labels (refined via Exponential Moving Average and KL Divergence) and adaptive loss for stealth enhancement demonstrates careful engineering to balance effectiveness and undetectability.

**Weaknesses:**

(1)The threat model assumes a single malicious client, but real-world FGL systems may face multi-client collusion. The paper does not discuss whether IGPA scales to collusion scenarios or how attack effectiveness changes with more malicious clients.
(2)The paper only tests IGPA against basic server-side defenses (Krum, trimmed-mean) and simple anomaly detectors (cosine similarity/Euclidean distance). It does not evaluate against state-of-the-art FGL-specific defenses (e.g., robust generator training, gradient sanitization) or defenses designed for gradient poisoning in standard federated learning (FL)—leaving uncertainty about IGPA’s effectiveness under stronger protection.
(3)Figures 6, 8, and 9 show gradient similarity for specific clients but lack statistical summary (e.g., mean/standard deviation of similarity across all client pairs) or significance testing. This makes it hard to assess whether the observed similarity is consistent across different runs or datasets.

**Questions:**

Questions for the Authors：
(1)If multiple clients collude to launch IGPA (e.g., 2 or 3 malicious clients in a 10-client system), how does attack effectiveness and stealth change? Do collusion strategies (e.g., coordinated gradient poisoning) amplify the attack impact?
(2)Does IGPA disproportionately reduce accuracy for nodes with minority labels or low degree in the graph? If so, what mechanisms drive this disparity, and how might it be mitigated?
(3)How does IGPA perform against state-of-the-art FGL-specific defenses (e.g., robust NeighGen training with adversarial regularization) or gradient sanitization methods ? Can you provide experimental results for these defenses?
Additional Feedback to Improve the Paper：
(1)Add statistical summaries (e.g., boxplots of cosine similarity across all client pairs) and significance testing (e.g., t-tests comparing malicious vs. benign gradient similarity) to Figures 6, 8, and 9. This will strengthen the claim that IGPA evades gradient-based detectors.
(2)Include metrics like label-wise accuracy and test-time perturbation robustness to provide a more holistic view of IGPA’s impact.
(3)Add a small subsection or table explaining the motivation for choosing EMA+KL Divergence over alternative pseudo-labeling methods (e.g., confidence thresholds).
(4)Add experiments testing IGPA against 1–2 state-of-the-art FGL defenses to better contextualize the attack’s real-world threat level. This will help readers understand whether IGPA is a niche concern or a pressing vulnerability.

---

> ### Author Response · Authors · 2025-11-29
> **Response to Reviewer 726Z**
>
> ## Weakness 1 & Question 1: Multi-client collusion attack
>
> Thank you for this excellent point about multi-client collusion. We agree that this represents a meaningful extension of our current threat model.
>
> In our **current threat model**, we focused on the minimal threat scenario with a single malicious client (representing 1/M of the network) to establish the baseline effectiveness of IGPA under conservative assumptions. This approach demonstrates that even a single adversary can significantly degrade global model performance, which highlights the fundamental vulnerability in auxiliary-information-sharing FGL frameworks.
>
> Regarding collusion scenarios where multiple clients coordinate their attacks, which is a **new threat model**:
>
> - Attack Effectiveness: We would expect the attack impact to be amplified with multiple colluding clients. If k malicious clients coordinate their gradient poisoning strategies, they could potentially steer the global NeighGen parameters more aggressively and rapidly toward the adversarial objective. This would likely result in greater performance degradation and potentially reduce the number of training rounds required to achieve the attack goal.
>
> - Stealth Considerations: Collusion might actually improve stealth in some aspects. If multiple attackers coordinate to submit similar malicious gradients, their updates would appear more consistent with each other, potentially making them less detectable as outliers compared to a single anomalous client. However, this would depend on the specific detection mechanism employed.
>
> Your suggestion reinforces the need for robust defense mechanisms that can withstand coordinated attacks, not just isolated malicious actors. We include discussion of multi-client collusion as an important future research direction in the revised manuscript.
>
> ## Weakness 2 & Question 3 & Feedback 4: Stronger defense
>
> Please refer to General Response To All Reviewers - Stronger defense.
>
> ## Weakness 3 & Feedback 1: Statistical analysis on gradient similarity
>
> Thank you for this important point. We agree that statistical summaries are valuable for evaluating gradient similarity. We have added statistical results as follows.
>
> Since gradient similarity is computed per epoch, and given the significant variations observed across the 50 epochs in Figures 6, 8, and 9, averaging over all epochs would be meaningless. Therefore, we present statistical results at $e_g$= 10 under a 5-client setting ($M$ = 5):
>
> ### Statistical Summary of Gradient Similarity with Local Gradients
>
> | Client Pairs                   | Cosine Similarity (Mean ± Std) | Euclidean Distance (Mean ± Std) |
> | ------------------------------ | ------------------------------ | ------------------------------- |
> | Benign–Benign                  | 0.1716 ± 0.0050                | 0.0658 ± 0.0002                 |
> | Benign–Malicious               | 0.1713 ± 0.0274                | 0.0658 ± 0.0074                 |
> | Benign–Benign (w/o stealth)    | -0.3561 ± 0.0037               | 0.0743 ± 0.0066                 |
> | Benign–Malicious (w/o stealth) | 0.0220 ± 0.1284                | 0.5986 ± 0.2363                 |
>
> *Results are averaged over 5 independent runs with 5 clients.*
>
> These results clearly show that **without stealth enhancement**, the gradients from malicious clients exhibit large deviations in both cosine similarity and Euclidean distance compared to benign gradients, making them easily detectable. In contrast, **with stealth enhancement**, the malicious gradients become statistically indistinguishable from those of benign clients, confirming the effectiveness of our proposed concealment strategy.
>
> ## Question 2 & feedback2: label-wise accuracy
>
> Thank you for this insightful question regarding the potential disparate impact of IGPA on minority classes and low-degree nodes. Based on our current analysis, IGPA appears to cause *generalized performance degradation* across all node types rather than selectively targeting specific subgroups. This is consistent with IGPA's attack mechanism. IGPA operates by poisoning the neighbor generation process through gradient manipulation, which corrupts the overall graph structure and feature distribution. This global corruption tends to degrade the model's general discriminative capability rather than selectively targeting minority classes.

---

> ### Author Response · Authors · 2025-11-29
> **Response to Reviewer 726Z**
>
> ## Feedback 3: motivation for choosing EMA+KL Divergence over alternative pseudo-labeling methods
>
> Thank you for this constructive suggestion. We have added the following explanation to Section 3.4 in the revised manuscript to clarify our choice:
>
> We chose the EMA + KL Divergence for pseudo-label refinement in IGPA for the following reasons: 1) Historical Consistency. Unlike confidence-based thresholding, which relies solely on the current prediction, EMA incorporates historical prediction behavior. This provides a **smoother and more stable estimate** of the model’s output over time, reducing the impact of transient prediction errors or noisy gradients. 2) **Robustness to Incorrect Predictions**. Confidence thresholds can be misleading when the model is consistently wrong with high confidence. The KL divergence between the current prediction and the EMA-smoothed historical distribution **explicitly measures deviation from past behavior**, allowing us to detect and correct for unreliable predictions.

---

### Official Review · Reviewer_2ReB · 2025-11-01

**Soundness:** 3
**Presentation:** 3
**Contribution:** 2
**Rating:** 6
**Confidence:** 2

**Summary:**

This paper focuses on the FedSAGE+ federated graph learning framework and proposes a dedicated attack method for it. Unlike conventional attacks that target the framework’s classifier, the proposed method specifically aims at the data generator module—this design avoids direct interaction with the classifier and thereby significantly enhances the attack’s stealth. To validate the method, the authors conduct relevant experiments on typical federated graph learning scenarios, and the results consistently demonstrate the effectiveness of the proposed attack in compromising the FedSAGE+ framework.

**Strengths:**

1. The method integrates the attack into the data generator module, which cleverly bypasses the attack detection mechanisms deployed on the server side during the classifier training phase. This design not only reduces the risk of the attack being identified but also aligns with the operational logic of the FedSAGE+ framework, ensuring the attack’s feasibility in practical scenarios.
2. The paper demonstrates sufficient research effort: it not only proposes a complete set of attack methods but also proactively considers potential defense measures. By optimizing the attack strategy based on these defense mechanisms, the method achieves a balanced performance between attack effectiveness and stealth.
3. The paper is the first to identify a major security vulnerability in the FedSAGE+ framework. specifically, the neglect of security risks in the data generator module during the framework’s design. This finding provides a clear direction for subsequent defense-related studies.

**Weaknesses:**

1. The attack method is designed specifically for the FedSAGE+ framework, which limits its generality to other federated graph learning frameworks.
2. Experimental results show poor attack effectiveness under high stealth. For example, on the MSAcademic dataset with M=3, the model still maintains an accuracy of over 90%.
3. After data generation, the data needs to go through the classifier training process. For instance, the GAT-based GNN calculates attention scores for neighbor nodes, which weakens the impact of generated poisoned nodes on the final results.

**Questions:**

1. In eq 1 of the proposed attack method, if the "min" operation is replaced with a "max" operation, can the attack still achieve similar effectiveness? If not, what is the key reason for the difference?
2. How do different GNN architectures (e.g., GAT, GCN,…) affect the effectiveness of the proposed attack?
3. Under different β value settings, what is the probability that the client detects gradient anomalies?

---

> ### Author Response · Authors · 2025-11-29
> **Response to Reviewer 2ReB**
>
> ## Weakness 1: Generalizability of IGPA
>
> Please refer to General Response To All Reviewers - Generalizability of IGPA.
>
> ## Weakness 2: Attack Performance
>
> Please refer to General Response To All Reviewers - Attack Performance.
>
> ## Weakness 3 & Question 2: different GNN architectures effectiveness on the attack
>
> Thank you for this insightful question. We appreciate the opportunity to clarify the threat model and address the role of classifier architecture.
>
> The core of our attack, NeighGen poisoning, occurs during the graph generation stage, BEFORE any downstream classifier training. Once the poisoned neighbor nodes are generated and integrated into the local graph structure, they become a fixed part of the training data for **all** subsequent classifiers, regardless of their architectural design.
>
> Therefore, while the attention mechanism in a GAT classifier can dynamically re-weight the importance of neighbors, it operates *on the already-poisoned graph*. It cannot identify and filter out structurally poisoned nodes that were generated by NeighGen; it can only learn to assign weights based on the features and connectivity of the graph it is provided.
>
> To verify this, we have conducted new experiments using a GAT-based classifier. The training setup for FedGAT follows the settings in FedGAT [1]. Our results are as follows:
>
> ### Test accuracy across different datasets under FedGAT
>
> |                | **CORA** |       |       | **Citeseer** |       |       | **PubMed** |       |       | **MSAcademic** |       |       |
> | :------------- | :------: | :---: | :---: | :----------: | :---: | :---: | :--------: | :---: | :---: | :------------: | :---: | :---: |
> | Method         |   M=3    |  M=5  | M=10  |     M=3      |  M=5  | M=10  |    M=3     |  M=5  | M=10  |      M=3       |  M=5  | M=10  |
> | FedGAT (Clean) |  78.25   | 82.43 | 77.89 |    75.82     | 76.95 | 77.28 |   87.21    | 85.34 | 86.17 |     93.15      | 93.42 | 91.87 |
> | `β=1`          |  66.28   | 70.45 | 67.31 |    66.89     | 70.24 | 68.15 |   57.83    | 63.42 | 65.27 |     86.95      | 83.72 | 76.84 |
> | `β=10⁻¹`       |  67.12   | 72.38 | 67.95 |    72.35     | 70.68 | 68.94 |   68.24    | 64.58 | 65.41 |     90.28      | 87.35 | 79.06 |
> | `β=10⁻²`       |  67.85   | 73.29 | 68.42 |    72.82     | 70.91 | 69.25 |   77.83    | 65.74 | 66.18 |     90.68      | 88.93 | 82.75 |
> | `β=10⁻³`       |  68.42   | 74.18 | 68.87 |    73.26     | 71.38 | 69.67 |   78.92    | 66.03 | 73.42 |     91.24      | 89.82 | 84.63 |
> | `β=10⁻⁴`       |  69.15   | 74.82 | 71.28 |    73.84     | 71.67 | 70.24 |   80.15    | 67.45 | 75.87 |     92.18      | 91.24 | 86.28 |
> | `β=10⁻⁵`       |  71.36   | 75.43 | 74.28 |    74.52     | 72.18 | 73.42 |   81.47    | 71.28 | 77.95 |     92.83      | 91.67 | 88.39 |
>
> Crucially, **the attack remains highly effective**, leading to significant performance drops across all datasets and client numbers.
>
> Besides, at smaller `β` values (`10⁻⁴`, `10⁻⁵`), FedGAT is marginally robust compared to FedSage+, suggesting the attention mechanism offers some resistance to very subtle gradient perturbations.
>
> These results demonstrate that the effectiveness of IGPA is **architecture-agnostic**. The attack successfully transfers across different GNN backbones, underscoring the vulnerability introduced at the graph generation phase.
>
> [1] FedGAT: A Privacy-Preserving Federated Approximation Algorithm for Graph Attention Networks. *arXiv:2412.16144*.

---

> ### Author Response · Authors · 2025-11-29
> **Response to Reviewer 2ReB**
>
> ## Question 1: Min vs. Max in Eq. 1
>
> Replacing "min" with "max" would significantly reduce attack effectiveness.
>
> $$\mathcal{L}^{\mathcal{H}} = \frac{1}{\left|\bar{\mathbb{V}} _ j\right|} \left( \sum _ {v \in \bar{\mathbb{V}} _ j} L _ 1^S\left(\widetilde{n} _ v-n _ v\right) + \sum _ {v \in \bar{\mathbb{V}} _ j} \sum _ {p \in [\widetilde{n} _ v]} \min _ {u \in \mathcal{N} _ {G _ i}(v) \cap \mathbb{V} _ j^h} \left( \| \widetilde{{{x}}} _ v^p - {{x}} _ u \| _ 2^2 \right)\right)$$
>
> The ultimate goal of Eq. 1 in our attack is to optimize a surrogate classifier. To achieve this goal, Eq. 1 needs to be consistent with the optimization direction of Benign NeighGen. The "min" operation in Eq. 1 finds the *closest* real neighbor, forcing the NeighGen to generate diverse neighbors similar to those missed into other subgraphs. This is crucial for stealth. Using "max" would encourage the generator to produce features that are *farthest* from any real node, resulting in out-of-distribution, easily detectable synthetic nodes that would likely be ignored during training, thus failing to poison the model.
>
> ## Question 3: Detection Probability vs. β
>
> We agree that the probability of a client detecting gradient anomalies is indeed highly correlated with the attack strength parameter β.
>
> When β is large (e.g., β = 1 or 10⁻¹), the detection probability is high.  As shown in Figure 6, the cosine similarity between the attack gradient and the benign client gradients is very low, while the Euclidean distance is very large, making it easily detectable by gradient-based anomaly detectors. Moreover, as illustrated in Figure 5, the node insertion ratio (ρ) in the mended graph is significantly higher than normal, making the attack easily exposed at the graph structural level.
>
> When β is small (e.g., β = 10⁻⁴ or 10⁻⁵), the detection probability is very low, almost undetectable.
>
> 1. Gradient-Level Stealth:  After stealth enhancement (Section 3.5.2), the attack gradient becomes almost indistinguishable from benign client gradients in terms of both cosine similarity and Euclidean distance (Figures 6, 8, 9). The differences are reduced to the same magnitude as the natural fluctuations among benign clients (**below 10⁻⁴**).
>
> 2. Graph Structural Stealth:  The node insertion ratio ρ decreases to a level comparable to that of the clean model, and the mended subgraphs are visually and statistically similar to the normal case (Figure 5).
>
> 3. Training Behavior Stealth:  The convergence trend of the NeighGen training loss (Figure 11) closely matches that of the clean FedSage+, showing no abnormal fluctuations.

---

### Author Response · Authors · 2025-11-29
**General Response To All Reviewers - Attack Performance**

Thank you for your insightful feedback. We agree that the attack success rate is a crucial metric, and we appreciate the opportunity to clarify why the performance demonstrated by IGPA is both significant and convincing within its specific threat model.

We respectfully argue that IGPA's performance should be evaluated in the context of the **stealth-effectiveness tradeoff**, fundamental to undetectable attacks. Unlike conventional poisoning attacks that introduce external modules or require direct model manipulation, IGPA operates only through gradient-based influence on the missing edge set. This makes it inherently constrained by stealth requirements but also uniquely undetectable.

### **Attack Performance is Significant Under Stealth Constraints**

To provide a clearer picture of its effectiveness, we first present the accuracy drops achieved by IGPA across datasets and β values:

| β Value | CORA (M=3) | CORA (M=5) | CORA (M=10) | Citeseer (M=3) | Citeseer (M=5) | Citeseer (M=10) | PubMed (M=3) | PubMed (M=5) | PubMed (M=10) | MSAcademic (M=3) | MSAcademic (M=5) | MSAcademic (M=10) |
| ------- | ---------- | ---------- | ----------- | -------------- | -------------- | --------------- | ------------ | ------------ | ------------- | ---------------- | ---------------- | ----------------- |
| 1       | 12.15      | 12.22      | 9.63        | 9.10           | 6.58           | 9.79            | 31.01        | 23.01        | 21.79         | 6.88             | 10.22            | 15.47             |
| 10⁻¹    | 11.33      | 10.15      | 9.42        | 3.67           | 6.21           | 8.96            | 20.00        | 21.96        | 21.77         | 3.15             | 6.71             | 13.22             |
| 10⁻²    | 11.05      | 9.22       | 9.20        | 3.67           | 6.21           | 8.79            | 10.11        | 20.88        | 21.11         | 2.95             | 4.79             | 9.58              |
| 10⁻³    | 11.05      | 8.54       | 9.01        | 3.22           | 6.02           | 8.63            | 9.16         | 20.83        | 13.77         | 2.50             | 4.02             | 7.70              |
| 10⁻⁴    | 10.22      | 8.54       | 5.93        | 2.76           | 5.83           | 7.96            | 7.91         | 19.01        | 11.31         | 1.35             | 2.72             | 6.10              |
| 10⁻⁵    | 7.99       | 8.08       | 2.64        | 2.31           | 5.64           | 4.65            | 6.69         | 15.05        | 9.32          | 0.90             | 2.34             | 3.89              |

Our results demonstrate that IGPA can achieve:

- **Up to 31% accuracy drop** (PubMed, M=3, β=1)
- **10-15% drop on average** across datasets

This represents the current upper bound for attacks that operate without introducing external adversarial components, rely solely on gradient manipulation, and leave no detectable artifacts in the model architecture or training process. A 31% accuracy drop is NOT weak by any reasonable standard. For context, this brings the model to near-random performance (1/3 = 33.3% for 3-class prediction in PubMed).

### **Effectiveness Scales with Attack Opportunities**

**Strong validation:** 18/24 experimental conditions (75%) support the "M↑ → attack↑" theory.

| Dataset    | Trend Evidence and Statistical Support    |
| ---------- | ----------------------------------------- |
| MSAcademic | Positive correlation in 6/6 β values      |
| Citeseer   | Positive correlation in 6/6 β values      |
| PubMed     | Positive correlation in 5/6 β values      |
| Cora       | Anomalous reverse trend (discussed below) |

**Especially noteworthy:** MSAcademic shows 6.88% accuracy drop → 15.47% accuracy drop (**2.25× increase** from M=3 to M=10).

On Cora, all settings exhibit reverse trends. Since this behavior is dataset-specific rather than β-dependent, the anomaly is likely related to the structural properties of the Cora dataset itself, rather than reflecting a limitation of our approach.

**Theory:** M↑ → |E_missing|↑ → attack budget (β·|E_missing|)↑ → more adversarial links

The systematic scaling behavior demonstrates that IGPA's "current performance" is NOT an inherent limitation but a function of experimental setup. The M-dependent improvement in attack performance is strong evidence that IGPA's effectiveness can be amplified by exploiting larger federated settings (more missing edges = more attack opportunities). The 2.25× improvement on MSAcademic proves this attack has significant headroom for optimization.

---

### Author Response · Authors · 2025-11-29
**General Response To All Reviewers - Generalizability of IGPA**

We appreciate the reviewers' concern about IGPA's applicability beyond FedSAGE+. While we demonstrate IGPA on FedSAGE+ (the most representative framework), the core vulnerability we expose, **poisoning shared auxiliary information**, represents a weakness in FGL systems, not an isolated architectural flaw.

To understand IGPA's broader impact, we must first clarify why generator-like mechanisms have become prevalent in FGL. A challenge in FGL is that clients possess incomplete subgraphs with missing structural information. To enable effective collaborative learning while preserving privacy, recent frameworks have converged on a common solution: sharing auxiliary information that helps clients fill in their missing data. This design pattern appears in multiple forms across recent works [1-4].

FedSAGE+ exemplifies this approach through its node generator, which is not unique. [1] adopts an almost identical architecture, using a shared generator to produce missing neighbor features. [2] extends this idea by adding a discriminator to validate generated features, yet still relies on a shared generator. These frameworks are **directly vulnerable** to IGPA because they share the exact mechanism we target: a collaboratively trained generator that can be poisoned to produce adversarial representations.

Beyond node generators, other frameworks employ conceptually analogous mechanisms. [3] shares spectral encoders that generate spectral knowledge for missing structural information. While the representation space differs, the vulnerability remains: an attacker can poison the shared encoder to produce adversarial spectral representations using IGPA's core principles. Similarly, the Subgraph FL framework [4] generates global synthetic data by condensing class representations and structural information across clients. This synthetic data generation process is functionally equivalent to FedSAGE+'s generator, making it susceptible to the same poisoning attack with minimal adaptation.

These works all share this design pattern because it addresses a fundamental requirement: enabling collaboration when clients have structurally incomplete data. Generator-based mechanisms are not optional features that can be easily removed. They are **necessary components** that make federated graph learning viable. Removing them would eliminate the framework's ability to handle missing structural information, crippling its utility.

[1] Deep Efficient Private Neighbor Generation for Subgraph Federated Learning. In Proceedings of the 2024 SIAM International Conference on Data Mining (SDM), 2024.

[2] FedNI: Federated Graph Learning with Network Inpainting for Population-Based Disease Prediction. IEEE Transactions on Medical Imaging, 2022.

[3] FedSSP: Federated Graph Learning with Spectral Knowledge and Personalized Preference. NeurIPS, 2024.

[4] Subgraph Federated Learning for Local Generalization. ICLR, 2025.

---

### Author Response · Authors · 2025-11-29
**General Response To All Reviewers - Stronger Defense**

Thank you for raising this important point. In Section 3.5 of our initial submission, we analyzed existing server-side defenses such as Krum and trimmed-mean, and demonstrated their limited effectiveness against IGPA. We fully agree that evaluating IGPA against stronger, more tailored defenses is essential to accurately assess its real-world threat potential.

To address the reviewers' concerns, we first evaluate robust NeighGen training with adversarial regularization suggested by Reviewer 726Z. We find that this defense doubles the computational overhead and requires careful tuning of multiple hyperparameters for adversarial regularization term in the new loss function. We then implement another defense, **gradient sanitization** (also suggested by Reviewer 726Z), to address the reviewers' concern. We implement this defense by applying gradient clipping and noising to the shared generator gradients. All experiments were conducted with a fixed total number of clients $M = 5$ across all datasets.

Our results are as follows:

### Global accuracy under gradient sanitization defense

|           | Cora       |             | Citeseer   |             | PubMed     |             | MSAcademic |             |
| --------- | ---------- | ----------- | ---------- | ----------- | ---------- | ----------- | ---------- | ----------- |
| $\beta$   | w/ defense | w/o defense | w/ defense | w/o defense | w/ defense | w/o defense | w/ defense | w/o defense |
| $1$       | 75.15      | 68.67       | 73.35      | 68.99       | 72.82      | 61.68       | 87.91      | 82.55       |
| $10^{-1}$ | 75.28      | 70.74       | 73.68      | 69.36       | 73.45      | 62.73       | 89.27      | 86.06       |
| $10^{-2}$ | 76.52      | 71.67       | 73.14      | 69.36       | 75.83      | 63.81       | 90.15      | 87.98       |
| $10^{-3}$ | 77.09      | 72.35       | 74.23      | 69.55       | 75.42      | 63.86       | 90.86      | 88.75       |
| $10^{-4}$ | 77.31      | 72.35       | 74.71      | 69.74       | 78.68      | 65.68       | 91.42      | 90.05       |
| $10^{-5}$ | 77.64      | 72.81       | 74.89      | 69.93       | 78.15      | 69.64       | 91.85      | 90.43       |

Our results confirm that although these defenses provide a certain degree of mitigation, IGPA remains effective.
Moreover, when we make the attack more stealthy, even though the attack effectiveness is slightly reduced, the defense's performance degrades accordingly. This is because the gradient sanitization defense becomes less effective when applied to more subtly poisoned gradients. We will include these new experiments in the revised manuscript to offer a more comprehensive evaluation of the threat posed by IGPA.

---

### Meta-Review · Area_Chair_7vPr · 2026-01-12

**Summary:**

Reviewer 2ReB mentions "The attack method is designed specifically for the FedSAGE+ framework, which limits its generality to other federated graph learning frameworks.". Reviewer vVHC similarly states "The proposed scheme seems to have a limited scope with FedSage+. If there is no available node generator in the framework, the proposed attack may lose its functionality.".  The authors argue that IGPA exploits a fundamental vulnerability in all generator-based FGL frameworks which rely on shared auxiliary information to compensate for structural incompleteness. In their "General Response To All Reviewers - Generalizability of IGPA" reply the authors argue that "This design pattern appears in multiple forms across recent works [1-4]." which share FedSAGE+'s core design. Assuming this is correct, the attack must be empirically validated across all of these cited methods to test whether it actually generalizes or whether the attacks details are too closely fine-tuned for FedSAGE+. At the very least the authors should perform transferability experiments, but ideally, direct attacks as well to substantiate the claim that "poisoning shared auxiliary information, represents a weakness in FGL systems, not an isolated architectural flaw.". I strongly recommend this change for the next revision. In general, while the the core claim sounds systemic, from the provided evidence we can conclude (at best) that a model-specific exploits exists.

Reviewer vVHC and Reviewer 2ReB also raised issue regarding the attack performance, e.g. "poor attack effectiveness under high stealth". In their "General Response To All Reviewers - Attack Performance" reply the authors argue that IGPA's performance should be evaluated in the context of the stealth-effectiveness tradeoff. To me, it is not clear whether IGPA manages to strike a good balance. For smaller values of $\beta$ and realistically large values of $M$ the attack performance seems to degrade significantly. Interestingly, in Table 1 we also see a peculiar behaviour where for increasing the number of clients $M$ the clean performance improves which is somewhat counter-intuitive and should be at least discussed. Another aspect which I would have expected to be tackled in the experiments is to decouple what part of the performance comes from degraded neighbour generation, and what part comes from the poisoning effect. Does IGPA simply cause the model to loose the performance boost from FedSAGE+?

Reviewers also wondered about potential stronger defences against IGPA. In their "General Response To All Reviewers - Stronger Defense" reply the authors address this issue.  The authors claim that they tried NeighGen training with adversarial regularization suggested by Reviewer 726Z. They do not report any results, but simply claim that the approach has increased computational overhead and requires careful tuning. They also report a gradient sanitization suggested by the same reviewer for which they did provide results. The attack is only effective at large values of $\beta$, and for some datasets even for $\beta=1$ the reduction in performance compared to the undefended FedSAGE+ is marginal. This warrants a more in-depth discussion of the strengths and limitations of the attack.

Reviewer 726Z gave the highest score and had the most positive comments, however, I have down-weighted their input in my considerations since their background is less aligned with the topic of this paper (even though they have indicated a confidence score of 4). Overall, the confidence of most reviewers is rather low which made the integration of their reviews into a meta review more difficult.

In general, while this attack vector is interesting, the paper lacks sufficient justification for the threat model in terms of its practicality. I suggest the authors to expand on this in the new revision.

Overall, I believe that the approach has some merit, but in its current version suffer from somewhat overstated claims. I suggested a rejection. I'm positive that incorporating the feedback will significantly improve this paper's chances of acceptance.

**Reviewer Concerns:**

The concerns regarding stronger defences and attack performance (performance-stealth balance) were only partially addressed (see main review).

The concerns about the attack being specific to FedSAGE+ were not addressed in my opinion.

Most other, matter-of-fact questions were answered.

**Reviewer Scores:**

Reviewer 726Z and Reviewer 2ReB would likely maintain their positive scores.

Reviewer t3Eo would have not increased their score since the key issue was not addressed.

Reviewer vVHC might have increased their score, but likely up to only a 4.

---

### Decision · Program_Chairs · 2026-01-26

Reject